# Provable Offline Reinforcement Learning with Human Feedback

**Wenhao Zhan** [* 1]   **Masatoshi Uehara** [* 2]   **Nathan Kallus** [3]   **Jason D. Lee** [1]   **Wen Sun** [2]

## Abstract

In this paper, we investigate the problem of offline reinforcement learning with human feedback where feedback is available in the form of preference between trajectory pairs rather than explicit rewards. Our proposed algorithm consists of two main steps: (1) estimate the implicit reward using Maximum Likelihood Estimation (MLE) with general function approximation from offline data and (2) solve a distributionally robust planning problem over a confidence set around the MLE. We consider the general reward setting where the reward can be defined over the whole trajectory and provide a novel guarantee that allows us to learn any target policy with a polynomial number of samples, as long as the target policy is covered by the offline data. This guarantee is the first of its kind with general function approximation. To measure the coverage of the target policy, we introduce a new single-policy concentrability coefficient, which can be upper bounded by the per-trajectory concentrability coefficient. We also establish lower bounds that highlight the necessity of such concentrability and the difference from standard RL, where state-action-wise rewards are directly observed. We further extend and analyze our algorithm when the feedback is given over action pairs.

## 1 Introduction

In standard reinforcement learning (RL) setting, the agent learns to maximize an observed numerical reward signal. However, finding appropriate numerical rewards can often be challenging in practice, and getting rewards right significantly impacts the effectiveness of RL algorithms (Wirth et al., 2017). To address this challenge, preference-based RL with human feedback (RLHF) has emerged as a promising alternative (Christiano et al., 2017). In RLHF, the agent does not receive a numerical reward signal, but rather feedback from a human expert in the form of *preferences* for a state-action trajectory in given pairs of trajectories. RLHF has gained considerable attention across multiple application domains, including games (MacGlashan et al., 2017; Christiano et al., 2017; Warnell et al., 2018), large language models (Ziegler et al., 2019; Stiennon et al., 2020; Wu et al., 2021; Nakano et al., 2021; Ouyang et al., 2022; Glaese et al., 2022; Bai et al., 2022; Ramamurthy et al., 2022; Liu et al., 2023), and robot learning (Brown et al., 2019; Shin et al., 2023).

In this work, we focus on the problem of offline RLHF, where the learning process relies exclusively on pre-collected offline data without active interaction with the environment. Offline RL has gained significant attention in various applications where conducting real-time online experiments may be costly. In the context of RLHF, an offline setting is particularly relevant due to the high cost and latency associated with obtaining human feedback. One of the key challenges in offline RL is the limited coverage of available offline data. Since coverage of the entire state-action space is rarely feasible in practice (Chen and Jiang, 2019a), recent empirical and theoretical approaches to offline RL leverage pessimism so as to rely only on the coverage of one comparator policy (possibly the optimal one), *i.e.*, the so-called partial coverage condition (Yu et al., 2020; Kidambi et al., 2020; Rashidinejad et al., 2021a; Li et al., 2022a; Shi et al., 2022; Yin and Wang, 2021; Xie et al., 2021; Uehara and Sun, 2021; Zhan et al., 2022a). In the context of RLHF, it is also crucial to develop algorithms that work under the partial coverage condition.

Despite its significance, there are very few algorithms specifically designed for offline RLHF with strong statistical guarantees. In this work, we provide such algorithms and guarantees when preferences depend on unknown reward functions over trajectories. Notably, we consider general reward functions that can be defined over the whole trajectory rather than just state-action pairs. This is consistent with many practical settings in natural language processing. For instance, all benchmarks presented in RL4LM (Ramamurthy et al., 2022) use metrics defined over the entire trajectories.

---

[1]Department of Electrical and Computer Engineering, Princeton University, Princeton, USA [2]Department of Computer Science, Cornell University, Ithaca, USA [3]School of Operations Research and Information Engineering, Cornell University, Ithaca, USA. Correspondence to: Wen Sun <ws455@cornell.edu>.

*Interactive Learning with Implicit Human Feedback Workshop at ICML 2023*, Honolulu, Hawaii, USA. Copyright 2023 by the author(s).

Our main contributions can be summarized as follows:

- We propose a simple algorithm with general function approximation that consists of two main steps: (1) estimate the implicit reward using Maximum Likelihood Estimation (MLE) with general function approximation from offline data and (2) solve a distributionally robust planning problem over a confidence set around the MLE.

- We prove that our algorithm can effectively compete with a target policy as long as the offline data cover the target policy. Our analysis leverages a newly defined concentrability coefficient which is tailored to RLHF. As the concentrability coefficient differs from that in the standard RL setting where state-action-wise rewards are directly observed, we establish lower bounds that highlight the necessity of our partial coverage condition.

- We extend the algorithm to the setting where the transition kernel is unknown, where we not only construct confidence sets for the reward function but also for the system dynamics. Notably, even though the reward can be trajectory-wise, we only need to estimate the per-step transition dynamics to ensure efficient learning.

- We further extend our results to the action-based comparison model, where preferences are defined over individual actions instead of entire trajectories based on the advantage function of the optimal policy (Ramachandran and Amir, 2007; Zhu et al., 2023). In comparison to the case of the trajectory-wise comparison model, we can establish a partial coverage guarantee using a concentrability coefficient on pairs of state-action pairs rather than trajectories. In this scenario, our sample complexity only scales with a bound on the advantage function, which can be much smaller than a bound on per-trajectory rewards as shown in Ross et al. (2011); Agarwal et al. (2019).

## 2 Related Work

**Reinforcement learning from human feedback.** The closest work to ours is (Zhu et al., 2023), which also studies offline RLHF, but their algorithm and analysis are restricted to linear models. Our algorithm and analysis extend to general function approximation. Indeed, general classes such as neural networks are commonly employed in practice (Christiano et al., 2017; Abdelkareem et al., 2022). In the special case of linear rewards and preferences over trajectories, while our algorithms differ, our guarantees recover theirs. So, our guarantees are more general; see Remark 1. Moreover, they only consider the setting where the transition kernel is known, while our work can also handle unknown transitions. Finally, in the case of action-based preferences, Zhu et al. (2023) cannot provide guarantees with partial coverage, even under their restriction to linear models. We demonstrate how to achieve meaningful guarantees under partial coverage and a soft margin (Assumption 6).

Wirth et al. (2017) provide a survey of Preference-based RL (PbRL). PbRL has received considerable attention in theoretical RL (Yue et al., 2012; Novoseller et al., 2020; Xu et al., 2020; Pacchiano et al., 2021; Chen et al., 2022) but the focus is largely on online PbRL. To the best of our knowledge, Zhu et al. (2023) is the only previous work to provide theoretical guarantees for offline PbRL.

**Offline RL.** In offline RL, one of the most critical challenges is addressing the issue of insufficient coverage in the offline data. It is well-known that naive methods are unable to learn the optimal policy in such scenarios (Rashidinejad et al., 2021b). To tackle this problem, numerous algorithms have been proposed with theoretical guarantees (Liu et al., 2020; Kumar et al., 2020; Jin et al., 2021; Rashidinejad et al., 2021b; Uehara and Sun, 2021; Li et al., 2022b; Shi et al., 2022; Jin et al., 2020; Xie et al., 2021; Zhan et al., 2022a). The most relevant work is (Uehara and Sun, 2021), which focuses on offline model-based RL with general function approximation. However, their methods cannot be directly applied to RLHF since per-step rewards are not observable in our setting. Furthermore, even in the standard RL setting, the construction of confidence intervals differs between our approach and theirs. Another related paper is Cheng et al. (2022), which considers the general offline pessimistic RL framework in the standard setting and also subtracts a reference term in their algorithm, similar to ours. However, our motivations for such reference terms are quite different from theirs. Additional detailed comparisons are given in Section 4.1 and Remark 3.

## 3 Preliminaries

We first introduce our offline RLHF setting with general function approximation.

**Markov decision processes.** We consider an episodic time-inhomogeneous Markov Decision Process (MDP) denoted by $\mathcal{M}$, which consists of a state space $\mathcal{S}$, an action space $\mathcal{A}$, an initial state distribution $P_0^\star \in \Delta_{\mathcal{S}}$, and a horizon $H \in \mathbb{N}^+$. At each step $h \in [H-1]$, we use $P_h^\star : \mathcal{S} \times \mathcal{A} \to \Delta_{\mathcal{S}}$ to denote the ground truth transitions. The ground truth reward function for the entire trajectory is denoted by $r^\star : \mathcal{T} \to [0, r_{\max}]$, where $\mathcal{T} = (\mathcal{S} \times \mathcal{A})^H$ represents the set of all possible trajectories. Note that $r^\star$ is a trajectory-wise reward, which is more general than state-action-wise rewards commonly considered in standard RL, which is the special case where for some $\{r_h^\star\}_{h=1}^H$ we have $r^\star(\tau) = \sum_{h=1}^H r_h^\star(s_h, a_h)$ for a trajectory $\tau = (s_1, a_1, \cdots, s_H, a_H)$.

A history-dependent policy $\pi := \{\pi_h\}_{h=1}^H$ is characterized by $\pi_h : (\mathcal{S} \times \mathcal{A})^{h-1} \times \mathcal{S} \to \Delta_{\mathcal{A}}$, specifying the probability of selecting actions for the agent at each step $h \in [H]$ based on the entire history. We denote the set of all such history-

dependent policies as $\Pi_{\text{his}}$. Given a policy $\pi$, we define its expected reward with respect to a general reward function $r$ and initial and transition distributions $P = \{P_h\}_{h=0}^{H-1}$ as $J(\pi; r, P) := \mathbb{E}_{\tau \sim (\pi, P)}[r(\tau)]$. Here, $\mathbb{E}_{\tau \sim (\pi, P)}[\cdot]$ represents the expectation over the trajectory distribution when executing the policy $\pi$ under the transition $P$ starting from $P_0$. We use $\mathbb{E}_{\tau \sim \pi}[\cdot]$ or $\mathbb{E}_\pi[\cdot]$ to denote the special case when $P$ is the ground truth distribution $P^\star := \{P_h^\star\}_{h=0}^{H-1}$.

The optimal policy, denoted $\pi^\star$, is the policy that maximizes the expected reward with respect to the true reward $r^\star$ and system dynamics $P^\star$, *i.e.*, $\pi^\star := \arg\max_{\pi \in \Pi_{\text{his}}} J(\pi; r^\star, P^\star)$. As the true reward function $r^\star$ is dependent on the entire trajectory, the optimal policy $\pi^\star$ is generally history-dependent. Thus, designing offline RLHF algorithms that can learn history-dependent policies is crucial.

For any policy $\pi$, we can define its state-action visitation measure as follows: $d_h^\pi(s, a) = \mathbb{P}^{\pi, P^\star}(s_h = s, a_h = a), \forall h \in [H]$, where $\mathbb{P}^{\pi, P^\star}(\cdot)$ denotes the distribution of the trajectory when executing policy $\pi$ in $P^\star$. We will also use $d^\pi(\tau)$ to denote $\mathbb{P}^{\pi, P^\star}(\tau)$ for the whole trajectory $\tau$.

A policy is Markovian if at each step it depends solely on the current state. When the reward is state-action-wise and the policy is Markovian, we can define the associated V- and Q-functions as $V_h^\pi(s) = \mathbb{E}_\pi[\sum_{t=h}^H r_t^\star(s_t, a_t)|s_h = s], \forall h \in [H], Q_h^\pi(s, a) = \mathbb{E}_\pi[\sum_{t=h}^H r_t^\star(s_t, a_t)|s_h = s, a_h = a], \forall h \in [H]$. It is well-known that when the reward is state-action-wise, the optimal policy $\pi^\star$ is both Markovian and deterministic. Furthermore, we have $V_h^{\pi^\star}(s) = \sup_\pi V_h^\pi(s)$ and $Q_h^{\pi^\star}(s, a) = \sup_\pi Q_h^\pi(s, a)$ for all $h \in [H]$. For brevity, we will use $V^\star$ and $Q^\star$ to represent the optimal state-value function and Q-function, respectively. The advantage function of the optimal policy, denoted by $A^\star$, is defined to be $A_h^\star(s, a) = Q_h^\star(s, a) - V_h^\star(s)$ for all $h \in [H], s \in \mathcal{S}, A \in \mathcal{A}$.

**Offline reinforcement learning with human feedback.** We focus on the problem of offline RLHF in this work. Specifically, in the trajectory-based pairwise comparison setting, we are provided with an offline dataset $\mathcal{D} = \{\tau^{n,0}, \tau^{n,1}, o^n\}_{n=1}^N$, where $\tau^{n,0} = \{s_h^{n,0}, a_h^{n,0}\}_{h=1}^H$ and $\tau^{n,1} = \{s_h^{n,1}, a_h^{n,1}\}_{h=1}^H$ are i.i.d. sampled from the distributions $\mu_0$ and $\mu_1$, respectively, and $o^n \in \{0, 1\}$ indicates preference for $\tau^{n,1}$ over $\tau^{n,2}$. We assume it satisfies the following preference model:

**Assumption 1** (Preference-based model). *Given a pair of trajectories $(\tau^0, \tau^1)$, $o \in \{0, 1\}$ satisfies*

$$P(o = 1 \mid \tau_0, \tau_1) = P(\tau_1 \text{ is preferred over } \tau_0 \mid \tau_0, \tau_1)$$
$$= \Phi(r^\star(\tau_1) - r^\star(\tau_0)).$$

*where $\Phi : \mathbb{R} \to [0, 1]$ is a monotonically increasing link function.*

A commonly used link function is the sigmoid function $\sigma(x) = 1/\{1 + \exp(-x)\}$, leading to the Bradley-Terry-Luce (BTL) model (Christiano et al., 2017).

The objective of offline RLHF is to learn a high-quality policy $\widehat{\pi} \in \Pi_{\text{his}}$, *i.e.*, with $J(\pi_{\text{tar}}; r^\star, P^\star) - J(\widehat{\pi}; r^\star, P^\star) \leq \epsilon$ where $\pi_{\text{tar}}$ is a target policy we want to compete with (potentially $\pi^\star$).

**General function approximation.** In our paper, we estimate the reward $r^\star$ with general function approximation. We introduce a function class $\mathcal{G}_r$, such as linear functions or neural networks, to approximate the true reward. For each $r \in \mathcal{G}_r$ and trajectory pair $(\tau^0, \tau^1)$, we denote the induced preference model with respect to $r$ as $P_r(o|\tau^0, \tau^1)$, defined as

$$P_r(o = 1 \mid \tau^0, \tau^1) := \Phi(r(\tau^1) - r(\tau^0)). \tag{1}$$

We use bracketing numbers to measure the complexity of $\{P_r : r \in \mathcal{G}_r\}$.

**Definition 1** ($\epsilon$-bracketing number of preferences). *We say $(g^1, g^2)$ is an $\epsilon$-bracket if $g^1(\cdot \mid \tau^0, \tau^1) \leq g^2(\cdot \mid \tau^0, \tau^1)$ and $\|g^1(\cdot \mid \tau^0, \tau^1) - g^2(\cdot \mid \tau^0, \tau^1)\|_1 \leq \epsilon$ for all trajectory-pairs $(\tau^0, \tau^1)$. The $\epsilon$-bracketing number of $\mathcal{G}_r$, denoted by $\mathcal{N}_{\mathcal{G}_r}(\epsilon)$, is the minimal number of $\epsilon$-brackets $(g^{n,1}, g^{n,2})_{n=1}^N$ needed so that for any $r \in \mathcal{G}_r$ there is a bracket $i \in [N]$ containing it, meaning $g^{i,1}(\cdot|\tau^0, \tau^1) \leq P_r(\cdot|\tau^0, \tau^1) \leq g^{i,2}(\cdot|\tau^0, \tau^1)$ for all trajectory-pairs $(\tau^0, \tau^1)$.*

The $\epsilon$-bracket number is widely used in statistics (van de Geer, 2000) to study MLE and related M-estimates. One example for which we can bound the $\epsilon$-bracket number is linear rewards under the BTL model (Pacchiano et al., 2021; Zhu et al., 2023).

**Proposition 1.** *Suppose $\|\phi(\tau)\|_2 \leq R \ \forall \tau \in \mathcal{T}$, $\mathcal{G}_r \subseteq \{\tau \mapsto \langle \phi(\tau), \theta \rangle : \|\theta\|_2 \leq B\}$ for some featurization $\phi : \mathcal{T} \to \mathbb{R}^d$ and $B > 0$, and the link function is $\Phi(\cdot) = \sigma(\cdot)$. Then for any $\epsilon \leq 1$, $\log \mathcal{N}_{\mathcal{G}_r}(\epsilon) \leq \mathcal{O}(d \log \frac{BR}{\epsilon})$.*

The proof is deferred to Appendix A. To handle unknown transitions, we similarly use function classes $\{\mathcal{G}_{P_h}\}_{h=0}^{H-1}$ to approximate the transition probabilities $\{P_h^\star\}_{0=1}^{H-1}$. Similarly, we use $\mathcal{N}_{\mathcal{G}_{P_h}}(\epsilon)$ to denote the $\epsilon$-bracket number of $\mathcal{G}_{P_h}$. The formal definition is deferred to Appendix D.

# 4 Trajectory-Based Pairwise-Comparison with Known Transition

In this section, we present our algorithm and analyze the sample complexity for the trajectory-based pairwise-comparison setting when the ground truth transition $P^\star$ is known. In Sections 5 and 6, we will further explore the unknown transition setting and the action-based comparison setting.

## 4.1 Algorithm

Our proposed algorithm, `FREEHAND` described in Algorithm 1, consists of the following two steps.

**Confidence set construction via MLE.** We construct a confidence set for the ground truth reward from the implicit preference feedback. We achieve this by selecting reward models that nearly maximize the log-likelihood of observed data up to a slackness parameter $\zeta$. We will show that the result, $\mathcal{R}(\mathcal{D})$, approximates the following confidence set:

$$\mathcal{R}'(\mathcal{D}) := \{r \in \mathcal{G}_r : \mathbb{E}_{\tau_0 \sim \mu_0, \tau_1 \sim \mu_1}[|\{r(\tau_1) - r(\tau_0)\} \\ - \{r^*(\tau_1) - r^*(\tau_0)\}|^2] \leq \xi\}$$

for a certain $\xi$. Here the distance between $r$ and $r^*$ is measured using the total variation distance (*i.e.*, $\ell_1$ norm) of $r(\tau_1) - r(\tau_0)$ and $r^*(\tau_1) - r^*(\tau_0)$ over the offline data.

**Distributionally robust policy optimization.** After constructing the confidence set, we search for the policy that maximizes the policy value under the least favorable reward model, the $r \in \mathcal{R}(\mathcal{D})$ minimizing the policy value $J(\pi; r, P^*)$ minus $\mathbb{E}_{\tau \sim \mu_{\mathrm{ref}}}[r(\tau)]$, where $\mu_{\mathrm{ref}}$ is an arbitrary known reference trajectory distribution. It is generally recommended to set $\mu_{\mathrm{ref}}$ to $\mu_1$, as we will explain later, possibly a sample-average approximation thereof based on $\{\tau^{1,1}, \ldots, \tau^{N,1}\}$. By selecting the least favorable reward model instead of the MLE solution $\widehat{r}$, we penalize policies that are not well-covered by the offline data. The need for a reference policy arises because the approximated confidence set measures the uncertainty for reward difference between two trajectories ($r(\tau_1) - r(\tau_0)$), but it cannot measure the uncertainty of the reward of a single trajectory.

In the following, we compare our algorithm to existing works. (Zhu et al., 2023) consider a pessimistic offline RL algorithm for RLHF specialized to the linear reward class setting, while our `FREEHAND` can handle general function approximation. Specifically, they construct the confidence set using the feature-covariance-rotated $\ell_2$-ball around the MLE $\hat{\theta}$, where $\widehat{r}(\tau) = \langle \phi(\tau), \hat{\theta} \rangle$. In contrast, our confidence set is obtained directly from the log-likelihood objective and is generic. Uehara and Sun (2021) proposes a model-based pessimistic offline RL algorithm when we have access to rewards. The confidence set construction correspondingly differs significantly. Cheng et al. (2022) considers a general offline pessimistic RL framework. In their policy optimization step, they also subtract the value of a reference policy. This similarity is superficial, however, as the motivations are different. We subtract the value because we can only measure the difference between rewards of any two trajectories, while their motivation is to obtain a certain robustness result (their proposition 3).

**Algorithm 1 `FREEHAND`**: oFfline ReinforcemEnt lEarning with HumAN feeDback

1: **Input**: offline datset $\mathcal{D}$, slackness parameter $\zeta$, reference distribution $\mu_{\mathrm{ref}}$, true transition $P^\star$
2: **MLE**: compute $\widehat{r} = \mathrm{argmax}_{r \in \mathcal{G}_r} \sum_{n=1}^{N} \log P_r(o = o^n \mid \tau^{n,1}, \tau^{n,0})$
3: **Confidence set construction**: construct $\mathcal{R}(\mathcal{D}) = \{r \in \mathcal{G}_r : \sum_{n=1}^{N} \log P_r(o = o^n \mid \tau^{n,0}, \tau^{n,1}) \geq \sum_{n=1}^{N} \log P_{\widehat{r}}(o = o^n \mid \tau^{n,0}, \tau^{n,1}) - \zeta\}$.
4: **Distributionally robust planning**: return

$$\widehat{\pi} = \mathrm{argmax}_{\pi \in \Pi_{\mathrm{his}}} \min_{r \in \mathcal{R}(\mathcal{D})} (J(\pi; r, P^\star) - \mathbb{E}_{\tau \sim \mu_{\mathrm{ref}}}[r(\tau)]).$$

## 4.2 Analysis

To analyze the sample complexity of `FREEHAND`, we first quantify the discrepancy between the offline data $\mathcal{D}$ and the distribution induced by the target policy $\pi_{\mathrm{tar}}$.

**Definition 2** (concentrability coefficient for preference-based feedback)**.** *The concentrability coefficient w.r.t. a reward class $\mathcal{G}_r$, a target policy $\pi_{\mathrm{tar}}$, and a reference policy $\mu_{\mathrm{ref}}$ is defined as*

$$C_r(\mathcal{G}_r, \pi_{\mathrm{tar}}, \mu_{\mathrm{ref}}) := \max\Bigg\{0,$$
$$\sup_{r \in \mathcal{G}_r} \frac{\mathbb{E}_{\tau^0 \sim \pi_{\mathrm{tar}}, \tau^1 \sim \mu_{\mathrm{ref}}}[r^\star(\tau^0) - r^\star(\tau^1) - r(\tau^0) + r(\tau^1)]}{\sqrt{\mathbb{E}_{\tau^0 \sim \mu_0, \tau^1 \sim \mu_1}[|r^\star(\tau^0) - r^\star(\tau^1) - r(\tau^0) + r(\tau^1)|^2]}}\Bigg\}.$$

Note, when we choose $\mu_{\mathrm{ref}} = \mu_1$, by Jensen's inequality, the value of $C_r(\mathcal{G}_r, \pi_{\mathrm{tar}}, \mu_1)$ can always be upper bounded by the per-trajectory concentration coefficient: $C_r(\mathcal{G}_r, \pi_{\mathrm{tar}}, \mu_1) \leq \sqrt{C_{\mathrm{tr}}}$ for any $\mathcal{G}_r$, where $C_{\mathrm{tr}} := \max_{\tau \in \mathcal{T}} \frac{d^{\pi_{\mathrm{tar}}}(\tau)}{\mu_0(\tau)}$. Moreover, while $C_r(\mathcal{G}_r, \pi_{\mathrm{tar}}, \mu_1)$ becomes $\sqrt{C_{\mathrm{tr}}}$ in the worst case (*e.g.*, when $\mathcal{G}_r$ is the set of all functions mapping from $\mathcal{T}$ to $\mathbb{R}$), it can generally be much smaller. For example, when using linear models, it is a relative condition number, as explained in Appendix C. Finally, when $\mu_{\mathrm{ref}} = d^{\pi_{\mathrm{tar}}}$, our coefficient becomes 0. This implies that $C_r(\mathcal{G}_r, \pi_{\mathrm{tar}}, \mu_1)$ could be small when $\pi_{\mathrm{tar}}$ and $\mu_{\mathrm{ref}}$ are close. While the concept of concentrability coefficient has been used in offline RL with explicit reward feedback (Chen and Jiang, 2019b; Song et al., 2022), this property is unique when the feedback is in the form of preferences.

In our following PAC analysis, we further assume the reward class $\mathcal{G}_r$ is realizable and bounded.

**Assumption 2** (Realizability)**.** *We have $r^\star \in \mathcal{G}_r$.*

**Assumption 3** (Boundedness)**.** *We have $0 \leq r(\tau) \leq r_{\max}$ for all $r \in \mathcal{G}_r$ and $\tau \in \mathcal{T}$.*

**Theorem 1.** *For any $\delta \in (0,1]$, let $\zeta = c_{\mathrm{MLE}} \log(\mathcal{N}_{\mathcal{G}_r}(1/N)/\delta)$ where $c_{\mathrm{MLE}} > 0$ is a universal constant, then under Assumption 1,2 and 3, with probability $1 - \delta$, we have*

$$
J(\pi_{\mathrm{tar}}; r^\star, P^\star) - J(\widehat{\pi}; r^\star, P^\star)
$$
$$
\leq \sqrt{\frac{cC_r^2(\mathcal{G}_r, \pi_{\mathrm{tar}}, \mu_{\mathrm{ref}})\kappa^2 \log(\mathcal{N}_{\mathcal{G}_r}(1/N)/\delta)}{N}}, \quad (2)
$$

*where $c > 0$ is a universal constant and $\kappa = (\inf_{x \in [-r_{\max}, r_{\max}]} \Phi'(x))^{-1}$.*

Theorem 1 indicates that FREEHAND can learn an $\epsilon$-optimal policy compared to $\pi_{\mathrm{tar}}$ with a sample complexity of

$$
N = \widetilde{\mathcal{O}}\left( \frac{C_r^2(\mathcal{G}_r, \pi_{\mathrm{tar}}, \mu_{\mathrm{ref}})\kappa^2 \log(\mathcal{N}_{\mathcal{G}_r}(1/N)/\delta)}{\epsilon^2} \right).
$$

Next we provide a detailed explanation of this sample complexity. Firstly, $C_r(\mathcal{G}_r, \pi_{\mathrm{tar}}, \mu_{\mathrm{ref}})$ represents the extent to which the dataset $\mathcal{D}$ covers the target policy $\pi_{\mathrm{tar}}$. In our theorem, to obtain a non-vacuous PAC guarantee, we only require the dataset $\mathcal{D}$ to cover the target policy $\pi_{\mathrm{tar}}$ (i.e., $C_r(\mathcal{G}_r, \pi_{\mathrm{tar}}, \mu_{\mathrm{ref}}) < \infty$). The distributionally robust optimization step plays a crucial role in obtaining this guarantee under partial coverage. In particular, invoking the above-mentioned third property of $C_r(\mathcal{G}_r, \pi_{\mathrm{tar}}, \mu_{\mathrm{ref}})$, when setting $\pi_{\mathrm{tar}} = \mu_{\mathrm{ref}}$, (2) is reduced to

$$
J(\mu_{\mathrm{ref}}; r^\star, P^\star) \leq J(\widehat{\pi}; r^\star, P^\star) \quad (3)
$$

This encourages us to choose $\mu_{\mathrm{ref}} = \mu_1$ (or $\mu_0$) as it will allow us to ensure our performance is at least larger than the performance associated with the offline data.

Secondly, $\log(\mathcal{N}_{\mathcal{G}_r}(1/N))$ measures the complexity of the function class $\mathcal{G}_r$. For example, when using linear models, it takes $\tilde{O}(d)$. We refer the reader to van de Geer (2000) for bracketing number computations for more general classes. Thirdly, $\kappa$ represents the non-linearity of the link function $\Phi$, which determines the difficulty of estimating the reward from human preferences. This dependence on $\kappa$ is present in the existing literature of RLHF, both in online settings (Pacchiano et al., 2021; Chen et al., 2022) and offline settings (Zhu et al., 2023).

**Remark 1** (Comparison to Zhu et al. (2023)). *By specializing our result to the linear model, we recover the result in Zhu et al. (2023). Specifically, the bracketing number is calculated in Proposition 1, and $C_r(\mathcal{G}_r, \pi_{\mathrm{tar}}, \mu_{\mathrm{ref}})$ is reduced to a relative condition number. The details are deferred to Appendix C.*

**Remark 2.** *In practice, to compute $\mathbb{E}_{\tau \sim \mu_1}[r(\tau)]$ in the second step, we can use the sample average, with an additional cost of $\sqrt{\log(1/\delta)/N}$ in the suboptimality bound in Eq. (2).*

### 4.3 Discussion of the Concentrability Coefficient

In the worst-case scenario (*i.e.*, $\mathcal{G}_r$ is the set of all functions mapping from $\mathcal{T}$ to $\mathbb{R}$), the value of $C_r(\mathcal{G}_r, \pi_{\mathrm{tar}}, \mu_1)$ is reduced to to the per-trajectory concentrability coefficient $C_{\mathrm{tr}}$. The per-trajectory concentrability coefficient is generally larger than the per-step concentrability coefficient $C_{\mathrm{st}}$ commonly used in the general offline RL literature. Specifically, $C_{\mathrm{st}}$ is defined as

$$
C_{\mathrm{st}} := \max_{s,a,h} d_h^{\pi_{\mathrm{tar}}}(s,a)/\mu_{0,h}(s,a),
$$

where $\mu_{0,h}(s,a)$ represents the marginal distribution at step $h$. In this section, we show the dependence on the per-trajectory concentrability coefficient is necessary for our offline RLHF context. This is intuitively because our RLHF setting involves reward functions defined over trajectories, reflecting the fact that human feedback is also trajectory-based.

In the next proposition, we first show that the per-trajectory concentrability coefficient $C_{\mathrm{tr}}$ can be exponentially larger than the per-step concentrability coefficient $C_{\mathrm{st}}$.

**Proposition 2.** *For any $S \geq 1, A \geq 2, H \geq 1, C \geq 1$, there exists an MDP $\mathcal{M}$ with horizon $H$, a policy $\pi_{\mathrm{tar}}$ and a data distribution $\mu_0$ such that $|\mathcal{S}| = S, |\mathcal{A}| = A$ and $C_{\mathrm{st}} = C$ while $C_{\mathrm{tr}} = C^H$.*

Proposition 2 indicates that $C_{\mathrm{tr}}$ can be significantly larger than $C_{\mathrm{st}}$. A natural question arises as to whether we can obtain suboptimality guarantees using $C_{\mathrm{st}}$. Unfortunately, the following lower bounds reveal that the suboptimality can scale with $C_{\mathrm{st}}^{H-1}$ in the worst case:

**Theorem 2.** *Set $\pi_{\mathrm{tar}} = \pi^\star$. Then, for any $C > 1$ and $H \geq 1$, there exists a dataset distribution $\mu_1$ such that we have*

$$
\inf_{\widehat{\pi}} \sup_{(\mathcal{M}, \mu_0) \in \Theta_{\mathrm{st}}(C)} \mathbb{E}_{\mathcal{D}}[J(\pi^\star; r^\star, P^\star) - J(\widehat{\pi}; r^\star, P^\star)]
$$
$$
\gtrsim \min\left\{ C - 1, \sqrt{\frac{(\max\{C,2\})^{H-1}(C-1)}{N}} \right\},
$$

*where $\widehat{\pi}$ is any mesurable function of the data $\mathcal{D}$ (and knows the information of $\mu_1$). $\Theta_{\mathrm{st}}(C)$ is the set of all MDP, offline distribution $(\mathcal{M}, \mu_0)$ such that $C_{\mathrm{st}} \leq C$. Note $\mathbb{E}_{\mathcal{D}}$ is taken with respect to the randomness in $\mathcal{D}$.*

In addition, with similar hard instances constrcuted in Theorem 2, we can show that scaling with $C_{\mathrm{tr}}$ is necessary in our setting:

**Theorem 3.** *Set $\pi_{\mathrm{tar}} = \pi^\star$. Then for any $C > 1$ and $H \geq 1$, there exists a dataset distribution $\mu_1$ such that we*

*have*

$$\inf_{\widehat{\pi}} \sup_{(\mathcal{M},\mu_0)\in\Theta_{\mathrm{tr}}(C)} \mathbb{E}_{\mathcal{D}}[J(\pi^{\star};r^{\star},P^{\star}) - J(\widehat{\pi};r^{\star},P^{\star})]$$

$$\gtrsim \min\left\{C-1, \sqrt{\frac{C-1}{N}}\right\},$$

*where $\widehat{\pi}$ is any mesurable function of the data $\mathcal{D}$ (and knows the information of $\mu_1$). $\Theta_{\mathrm{tr}}$ is the set of all MDP, offline distribution $(\mathcal{M},\mu_0)$ such that $C_{\mathrm{tr}} \leq C$. Note $\mathbb{E}_{\mathcal{D}}$ is taken with respect to the randomness in $\mathcal{D}$.*

Note that when $\mu_1$ is known, we can set $\mu_{\mathrm{ref}} = \mu_1$ in Algorithm 1 and then $C_r(\mathcal{G}_r, \pi_{\mathrm{tar}}, \mu_1) \leq \sqrt{C_{\mathrm{tr}}}$, which implies the sample complexity in Theorem 1 indeed nearly matches this lower bound with respect to $C_{\mathrm{tr}}$ and $N$ when $N$ is sufficiently large.

In summary, Theorem 2 and Theorem 3 imply that the scaling with the per-trajectory concentrability coefficient is essential in the trajectory-based pairwise-comparison setting, and it cannot be relaxed to the per-step concentrability without additional assumptions, such as on the reward structure.

## 5 Trajectory-Based Comparison with Unknown Transition

We extend the setting presented in Section 4 to the scenario where the transition function $P^{\star}$ is unknown. The algorithm is described in Algorithm 2. Compared to Algorithm 1, we simply added a similar step to handle unknown transitions. Hereafter, we use the convention $P_0(\cdot \mid s,a) := P_0(\cdot)$.

Our sample complexity will depend on the following additional concentration coefficient:

**Definition 3** (Concentrability coefficient for the transition).
*The concentrability coefficient w.r.t. transition classes $\{\mathcal{G}_{P_h}\}$ and a target policy $\pi_{\mathrm{tar}}$ is defined as*

$$C_P(\{\mathcal{G}_{P_h}\}, \pi_{\mathrm{tar}}) := \max_{h:0\leq h\leq H-1} \sup_{P_h\in\mathcal{G}_{P_h}}$$

$$\frac{\mathbb{E}_{(s,a)\sim d_h^{\pi_{\mathrm{tar}}}}[\|P_h(\cdot\mid s,a) - P_h^{\star}(\cdot\mid s,a)\|_1]}{\sqrt{\mathbb{E}_{(s,a)\sim(\mu_{0,h}/2+\mu_{1,h}/2)}[\|P_h(\cdot\mid s,a) - P_h^{\star}(\cdot\mid s,a)\|_1^2]}}.$$

Note this is always upper-bounded by the density-ratio-based concentrability coefficient, $C_P(\{\mathcal{G}_{P_h}\}, \pi_{\mathrm{tar}}) \leq \sup_{(s,a,h)\in\mathcal{S}\times\mathcal{A}\times[H]} \frac{d_h^{\pi_{\mathrm{tar}}}(s,a)}{\mu_{0,h}(s,a)/2+\mu_{1,h}(s,a)/2}$.

We also assume the transition classes $\{\mathcal{G}_{P_h}\}_{h=0}^{H-1}$ are realizable:

**Assumption 4** (Realizability). *Suppose that we have $P_h^{\star} \in \mathcal{G}_{P_h}$ for all $h$ where $0 \leq h \leq H-1$. In addition, any choice $P_h \in \mathcal{G}_{P_h}$ for $0 \leq h \leq H-1$ are valid transition distributions.*

---

**Algorithm 2 FREEHAND-transition**

**Input**: offline dataset $\mathcal{D}$, slackness parameter $\zeta, \zeta_{P_h}$, reference distribution $\mu_{\mathrm{ref}}$
**MLE for reward**: compute
$\widehat{r} = \arg\max_{r\in\mathcal{G}_r} \sum_{n=1}^N \log P_r(o = o^n|\tau^{n,1}, \tau^{n,0})$.
**MLE for transition**: compute
$\widehat{P}_h = \arg\max_{P_h\in\mathcal{G}_{P_h}} \sum_{n=1}^N \sum_{i=0}^1 \log P_h(s_{h+1}^{n,i}|s_h^{n,i}, a_h^{n,i})$.
**Confidence set construction**: for $0 \leq h \leq H-1$, construct

$$\mathcal{R}(\mathcal{D}) = \left\{r \in \mathcal{G}_r : \sum_{n=1}^N \log P_r(o = o^n|\tau^{n,0}, \tau^{n,1}) \geq \right.$$

$$\left. \sum_{n=1}^N \log P_{\widehat{r}}(o = o^n|\tau^{n,0}, \tau^{n,1}) - \zeta\right\},$$

$$\mathcal{P}_h(\mathcal{D}) =$$

$$\left\{P_h \in \mathcal{G}_{P_h} : \sum_{n=1}^N \sum_{i=0}^1 \log P_h(s_{h+1}^{n,i}|s_h^{n,i}, a_h^{n,i})\right.$$

$$\left. \geq \sum_{n=1}^N \sum_{i=0}^1 \log \widehat{P}_h(s_{h+1}^{n,i}|s_h^{n,i}, a_h^{n,i}) - \zeta_{P_h}\right\}.$$

**Distributionally robust plnanning**: return $\widehat{\pi} = \arg\max_{\pi\in\Pi_{\mathrm{his}}} \min_{r\in\mathcal{R}(\mathcal{D}), P_h\in\mathcal{P}_h(\mathcal{D})} J\left(\pi; r, \{P_h\}_{h=0}^{H-1}\right) - \mathbb{E}_{\tau\sim\mu_{\mathrm{ref}}}[r(\tau)]$.

---

Then with the above assumptions, we have the following theorem to characterize the sample complexity when the transition is unknown:

**Theorem 4.** *For any $\delta \in (0,1]$, let $\zeta = c_{\mathrm{MLE}} \log(\mathcal{N}_{\mathcal{G}_r}(1/N)/\delta), \zeta_{P_h} = c_P \log(H\mathcal{N}_{\mathcal{G}_{P_h}}(1/N)/\delta)$ where $c_{\mathrm{MLE}}, c_P > 0$ are universal constants, then under Assumption 1,2,3 and 4, we have*

$$J(\pi_{\mathrm{tar}}; r^{\star}, P^{\star}) - J(\widehat{\pi}; r^{\star}, P^{\star})$$

$$\leq \sqrt{\frac{cC_r^2(\mathcal{G}_r, \pi_{\mathrm{tar}}, \mu_{\mathrm{ref}})\kappa^2 \log(\mathcal{N}_{\mathcal{G}_r}(1/N)/\delta)}{N}}$$

$$+ Hr_{\max}\sqrt{\frac{cC_P^2(\{\mathcal{G}_{P_h}\}, \pi_{\mathrm{tar}}) \log(H\mathcal{N}_P(1/N)/\delta)}{N}},$$

*where $c > 0$ and $\kappa$ are the same as Theorem 1 and $\mathcal{N}_P := \max_{0\leq h\leq H-1} \mathcal{N}_{\mathcal{G}_{P_h}}$.*

Compared to Theorem 1, we introduce an additional term in our guarantee to account for the unknown transitions. Once again, our result demonstrates that the learned policy can achieve performance comparable to any target policy $\pi_{\mathrm{tar}}$ covered by the offline data, i.e., $C_r(\mathcal{G}_r, \pi_{\mathrm{tar}}, \mu_{\mathrm{ref}}) < \infty, C_P(\{\mathcal{G}_{P_h}\}, \pi_{\mathrm{tar}}) < \infty$.

**Remark 3** (Comparison to Uehara and Sun (2021)). *Like us, Uehara and Sun (2021) proposed a model-based RL algorithm that works under partial coverage, but in the standard RL setting and with a known state-action-wise reward function. In addition to the difference in settings, which is the primary difference, our approach moreover differs from their approach because while they construct*

*confidence intervals by defining a confidence ball around the MLE solution based on the total variation distance, we use the Kullback-Leibler (KL) distance. This may be preferable as computing the KL distance is generally easier than the total variation distance as it arises directly from the MLE objective, as practically done in Rigter et al. (2022).*

# 6  Action-Based Comparison

Next, we turn our attention to the action-based comparison setting (Ramachandran and Amir, 2007; Zhu et al., 2023), where human evaluators provide preferences between pairs of actions instead of pairs of trajectories. In this section, we assume that the reward function $r^\star$ is state-action-wise: $r^\star(\tau) = \sum_{h=1}^{H} r_h^\star(s_h, a_h)$ for $\tau = (s_1, a_1, \cdots, s_H, a_H)$. And, we consider a preference model based on $Q^\star$.

**Setting.** We have datasets $\mathcal{D} = \{\mathcal{D}_h\}_{h=1}^{H}$ with $\mathcal{D}_h = \{(s_h^n, a_h^{n,0}, a_h^{n,1}, o_h^n)\}_{n=1}^{N}$ for each $h \in [H]$, where each sample is drawn i.i.d. from the distribution $s_h^n \sim \mu_h, a_h^{n,0} \sim \mu_{0,h}(\cdot \mid s_h^n), a_h^{n,1} \sim \mu_{1,h}(\cdot \mid s_h^n)$ and $o_h^n \in \{0, 1\}$ indicates preference for $a_h^{1,n}$ over $a_h^{0,n}$ in the state $s_h^n$. We assume it satisfies the following preference model:

**Assumption 5** (Action-based comparison model)**.** *Given a pair of actions $a_h^0, a_h^1$ and state $s_h$, $o \in \{0, 1\}$ satisfies*

$$P(o_h = 1 \mid s_h, a_h^0, a_h^1) = \Phi(Q_h^\star(s_h, a_h^1) - Q_h^\star(s_h, a_h^0)).$$

Here, the aforementioned distribution can be equivalently expressed as $P(o_h^n = 1 \mid s_h^n, a_h^{n,0}, a_h^{n,1}) = \Phi(A_h^\star(s_h^n, a_h^{n,1}) - A_h^\star(s_h^n, a_h^{n,0}))$, where $A^\star$ denotes the optimal advantage function. Consequently, we introduce general function classes $\mathcal{G}_{A_h}$ to estimate the optimal advantage function $A_h^\star$. In addition, for each $A_h \in \mathcal{G}_{A_h}$ and $(s, a^0, a^1) \in \mathcal{S} \times \mathcal{A} \times \mathcal{A}$, we use $P_{A_h}(\cdot \mid s, a^0, a^1)$ to represent the human preference model with respect to $A_h$, defined as $P_{A_h}(o = 1 \mid s, a^0, a^1) := \Phi(A_h(s, a^1) - A_h(s, a^0))$.

We again use the $\epsilon$-bracket number of such advantage function classes to quantify their complexity, denoted as $\mathcal{N}_{\mathcal{G}_{A_h}}$. The full formal definition is provided in Appendix D.

## 6.1  Algorithm

Our algorithm comprises two steps. In the first step, our objective is to estimate the optimal advantage function using MLE. In the second step, we determine the policy by selecting the action with the highest advantage value based on the learned advantage function.

## 6.2  Analysis

Now we show that FREEHAND-action is able to learn a near-optimal policy as long as offline data covers the optimal

---

**Algorithm 3** `FREEHAND-action`

1: **Input**: offline datset $\mathcal{D}$.
2: **MLE**: compute $\widehat{A}_h = \text{argmax}_{A_h \in \mathcal{G}_{A_h}} \sum_{n=1}^{N} \log P_{A_h}(o = o_h^n \mid s_h^n, a_h^{n,0}, a_h^{n,1}), \forall h \in [H]$.
3: **Greedy policy**: return $\widehat{\pi}_h(s) = \text{argmax}_{a \in \mathcal{A}} \widehat{A}_h(s, a)$

---

policy. Our analysis depends on the following assumption on the margin of $Q^\star$:

**Assumption 6** (Soft margin)**.** *There exists $\alpha_0 \in \mathbb{R}^+$, $\beta \in (0, \infty]$ such that for all $a \in \mathcal{A}, h \in [H], \alpha > 0$, we have $\mathbb{P}^{\pi^\star, P^\star}(0 < |Q_h^\star(s_h, \pi^\star(s_h)) - Q_h^\star(s_h, a)| < \alpha) \leq (\alpha/\alpha_0)^\beta$.*

The soft margin is widely used in the literature on classification, decision making, and RL (Audibert and Tsybakov, 2007; Perchet and Rigollet, 2013; Luedtke and Chambaz, 2020; Hu et al., 2021; 2022; Uehara et al., 2023). Note, when the optimal Q function satisfies a gap (as in Simchowitz and Jamieson, 2019; Wu et al., 2022), the soft margin assumption holds with $\beta = \infty$.

Next, we introduce the concentrability coefficient for the action-based comparison setting, which is defined as follows.

**Definition 4** (concentrability coefficient for action-based comparison)**.**

$$C_{\text{act}} := \sup_{h \in [H], A_h \in \mathcal{G}_{A_h}}$$

$$\frac{\mathbb{E}_{(s, a^0) \sim d_h^{\pi^\star}, a^1 \sim \text{Unif}(\cdot \mid s)}[l(A_h, s, a^0, a^1)]}{\mathbb{E}_{s \sim \mu_h, a^0 \sim \mu_{0,h}(\cdot \mid s), a^1 \sim \mu_{1,h}(\cdot \mid s)}[l(A_h, s, a^0, a^1)]},$$

*where $l(A_h, s, a^0, a^1) := |A_h^\star(s, a^0) - A_h^\star(s, a^1) - A_h(s, a^0) + A_h(s, a^1)|^2$ and $\text{Unif}(\cdot \mid s)$ is the uniform policy over $\mathcal{A}$.*

We observe that

$$C_{\text{act}} \leq \left( \sup_{h \in [H], s \in \mathcal{S}} \frac{d_h^{\pi^\star}(s)}{\mu_h(s)} \right)$$

$$\cdot \left( \sup_{h \in [H], s \in \mathcal{S}, a^0 \in \mathcal{A}} \frac{\pi_h^\star(a^0 \mid s)}{\mu_{0,h}(a^0 \mid s)} \right)$$

$$\cdot \left( \frac{1}{|\mathcal{A}|} \sup_{h \in [H], s \in \mathcal{S}, a^1 \in \mathcal{A}} \frac{1}{\mu_{1,h}(a^1 \mid s)} \right).$$

Based on this bound, we can consider simple sufficient conditions for $C_{\text{act}}$ to be finite. Firstly, regarding the first term, it is sufficient for the dataset distribution $\mu_h$ to cover the states visited by the optimal policy $\pi^\star$, denoted as $d_h^{\pi^\star}$. Regarding the second term, we require $\mu_{0,h}$ to cover $\pi_h^\star$. Additionally, the third term can be upper bounded when $\mu_{1,h}$ can cover the whole action space. This is mild because $\forall s \in$

$\mathcal{S}$; $\mu_h(s) > 0$ is not controllable to the learner; but $\forall (s, a) \in \mathcal{S} \times \mathcal{A}$; $\mu_{1,h}(a \mid s) > 0$ is controllable to the learner in the data-collection process. To summarize, $C_{\text{act}} < \infty$ primarily requires partial coverage over the state space with respect to the optimal policy, which is preferable in practical applications where $\mathcal{S}$ can be very large.

Additionally, we introduce several assumptions on the function classes similar to those in Section 4.

**Assumption 7.** *For all $h \in [H]$, we have $A_h^\star \in \mathcal{G}_{A_h}$.*

**Assumption 8.** *For all $h \in [H]$ and $A_h \in \mathcal{G}_{A_h}$, we have $|A_h(s, a)| \le b_{\max}$ for all $(s, a) \in \mathcal{S} \times \mathcal{A}$.*

With the aforementioned assumptions, we can establish the sample complexity of FREEHAND-action.

**Theorem 5.** *Under Assumption 5,6,7 and 8, we have with probability at least $1 - \delta$ that*

$$J(\pi^\star; r^\star, P^\star) - J(\widehat{\pi}; r^\star, P^\star) \le cH|\mathcal{A}| \left(\frac{2}{\beta}\right)^{\frac{\beta-2}{\beta+2}} \left(\frac{1}{\alpha_0}\right)^{\frac{2\beta}{\beta+2}}$$
$$\cdot \left(\frac{\kappa_A^2 C_{\text{act}} \log(H\mathcal{N}_{\mathcal{G}_A}(1/N)/\delta)}{N}\right)^{\frac{\beta}{\beta+2}},$$

*where $\mathcal{N}_{\mathcal{G}_A} := \max_{h \in [H]} \mathcal{N}_{\mathcal{G}_{A_h}}$ and $\kappa_A = \frac{1}{\inf_{x \in [-b_{\max}, b_{\max}]} \Phi'(x)}$.*

Theorem 5 suggests that FREEHAND-action can learn a near-optimal policy as long as $C_{\text{act}}$ takes a finite value under a soft margin. Specifically, when a hard margin is imposed (i.e., $\beta = \infty$), FREEHAND-action can learn an $\epsilon$-optimal policy with a sample complexity of $N = \widetilde{\mathcal{O}}(1/\epsilon)$, which is faster than a typical rate $\widetilde{\mathcal{O}}(1/\epsilon^2)$. As mentioned earlier, the quantity $C_{\text{act}}$ represents the extent to which the distribution induced by the optimal policy is covered by the offline data. Therefore, there is no need for a potentially stringent condition that requires the offline data to cover the entire state space like Zhu et al. (2023).

Furthermore, our guarantee is designed to overcome the limitations of existing approaches. In Theorem 1, our upper-bound is influenced by the parameter $\kappa$. When using a common sigmoid link function, this parameter scales with $\Theta(\exp(r_{\max}))$. As a result, in dense reward settings where $r_{\max}$ scales with $H$, this scaling factor may lead to an explicit dependence of $\Theta(\exp(H))$. Similar observations have been made in previous works (Zhu et al., 2023; Pacchiano et al., 2021; Chen et al., 2022). However, even if $r_{\max}$ scales with $H$, it is known that the $\ell_\infty$-norm of the advantage function, denoted as $b_{\max}$, can take much smaller values (Ross et al., 2011; Agarwal et al., 2019) Hence, we can avoid the explicit dependence on $\Theta(\exp(H))$.

## 7 Conclusions

We propose the first algorithm for RLHF with preferences over trajectories with general function approximation and under partial coverage. We establish lower bounds that explain the differences between our RLHF model and standard RL with direct reward feedback. Moreover, we extend our algorithm to unknown transitions and to preference feedback over actions, all while maintaining strong guarantees under partial coverage.

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

# A  Proof of Proposition 1

Let $\mathcal{F}$ denote the function class $\{f_r : f_r(\tau^0, \tau^1) = P_r(o = 1|\tau^0, \tau^1), r \in \mathcal{G}_r\}$. Let $\mathcal{I}_{\mathcal{F}}(\epsilon)$ denote the $\epsilon$-bracket number with respect to $\ell_\infty$-norm, i.e., the minimum integer $M$ such that there exist $M$ functions $\{f^i\}_{i=1}^M$ such that for each $f_r \in \mathcal{F}$, we have $\sup_{\tau^0, \tau^1} |f_r(\tau^0, \tau^1) - f^i(\tau^0, \tau^1)| \leq \epsilon$ for some $i \in [M]$. Then we know there exists a set of function $\overline{\mathcal{F}}$ with $|\overline{\mathcal{F}}| = \mathcal{I}_{\mathcal{F}}(\epsilon/4)$ such that for each $f_r \in \mathcal{F}$, there exists $\overline{f} \in \overline{\mathcal{F}}$ satisfying

$$\sup_{\tau^0, \tau^1} |f_r(\tau^0, \tau^1) - \overline{f}(\tau^0, \tau^1)| \leq \epsilon/4.$$

Now we construct a bracket $(g_{\overline{f}}^1, g_{\overline{f}}^2)$ defined as follows:

$$g_{\overline{f}}^1(o = 1|\tau^0, \tau^1) = \overline{f}(\tau^0, \tau^1) - \epsilon/4,$$
$$g_{\overline{f}}^1(o = 0|\tau^0, \tau^1) = 1 - \overline{f}(\tau^0, \tau^1) - \epsilon/4,$$
$$g_{\overline{f}}^2(o = 1|\tau^0, \tau^1) = \overline{f}(\tau^0, \tau^1) + \epsilon/4,$$
$$g_{\overline{f}}^2(o = 0|\tau^0, \tau^1) = 1 - \overline{f}(\tau^0, \tau^1) + \epsilon/4.$$

Then clearly we have $g_{\overline{f}}^1(\cdot|\tau^0, \tau^1) \leq P_r(\cdot|\tau^0, \tau^1) \leq g_{\overline{f}}^2(\cdot|\tau^0, \tau^1)$ and $\|g_{\overline{f}}^1(\cdot|\tau^0, \tau^1) - g_{\overline{f}}^2(\cdot|\tau^0, \tau^1)\|_1 \leq \epsilon$. This implies that $\mathcal{N}_{\mathcal{G}_r}(\epsilon) \leq \mathcal{I}_{\mathcal{F}}(\epsilon/4)$.

Now we only need to bound $\mathcal{I}_{\mathcal{F}}(\epsilon/4)$. Consider $\theta$ and $\theta'$ with $\|\theta - \theta'\|_2 \leq \epsilon_1$ and let $r$ ($r'$) denote the reward $\langle \phi, \theta \rangle$ ($\langle \phi, \theta' \rangle$). Then we know for all $\tau$,

$$|r(\tau) - r'(\tau)| \leq R\epsilon_1.$$

Fix the trajectory pair $(\tau^0, \tau^1)$. Without loss of generality, we assume $\exp(r(\tau^0)) + \exp(r(\tau^1)) \leq \exp(r'(\tau^0)) + \exp(r'(\tau^1))$. Then we have

$$\exp(r(\tau^0)) + \exp(r(\tau^1)) \leq \exp(r'(\tau^0)) + \exp(r'(\tau^1))$$
$$\leq \exp(R\epsilon_1)\Big( \exp(r(\tau^0)) + \exp(r(\tau^1)) \Big).$$

On the other hand, we have

$$|f_r(\tau^0, \tau^1) - f_{r'}(\tau^0, \tau^1)|$$
$$= \frac{1}{\Big( \exp(r'(\tau^0)) + \exp(r'(\tau^1)) \Big)}$$
$$\cdot \frac{1}{\Big( \exp(r(\tau^0)) + \exp(r(\tau^1)) \Big)}$$
$$\cdot \Big| \exp(r(\tau^1))\Big( \exp(r'(\tau^0)) + \exp(r'(\tau^1)) \Big)$$
$$- \exp(r'(\tau^1))\Big( \exp(r(\tau^0)) + \exp(r(\tau^1)) \Big)\Big|.$$

Therefore, if $\exp(r(\tau^1))\Big( \exp(r'(\tau^0)) + \exp(r'(\tau^1)) \Big) - \exp(r'(\tau^1))\Big( \exp(r(\tau^0)) + \exp(r(\tau^1)) \Big) \geq 0$, then we have

$$\Big| \exp(r(\tau^1))\Big( \exp(r'(\tau^0)) + \exp(r'(\tau^1)) \Big)$$
$$- \exp(r'(\tau^1))\Big( \exp(r(\tau^0)) + \exp(r(\tau^1)) \Big)\Big|$$
$$\leq \exp(R\epsilon_1) \exp(r(\tau^1))\Big( \exp(r(\tau^0)) + \exp(r(\tau^1)) \Big)$$
$$- \exp(-R\epsilon_1) \exp(r(\tau^1))\Big( \exp(r(\tau^0)) + \exp(r(\tau^1)) \Big)$$
$$= (\exp(R\epsilon_1) - \exp(-R\epsilon_1)) \exp(r(\tau^1))$$
$$\cdot \Big( \exp(r(\tau^0)) + \exp(r(\tau^1)) \Big).$$

Otherwise, we have

$$\Big| \exp(r(\tau^1))\Big( \exp(r'(\tau^0)) + \exp(r'(\tau^1)) \Big)$$
$$- \exp(r'(\tau^1))\Big( \exp(r(\tau^0)) + \exp(r(\tau^1)) \Big)\Big|$$
$$\leq \exp(R\epsilon_1) \exp(r(\tau^1))\Big( \exp(r(\tau^0)) + \exp(r(\tau^1)) \Big)$$
$$- \exp(r(\tau^1))\Big( \exp(r(\tau^0)) + \exp(r(\tau^1)) \Big)$$
$$= (\exp(R\epsilon_1) - 1) \exp(r(\tau^1))\Big( \exp(r(\tau^0)) + \exp(r(\tau^1)) \Big).$$

Therefore we have

$$|f_r(\tau^0, \tau^1) - f_{r'}(\tau^0, \tau^1)|$$
$$\leq \frac{(\exp(R\epsilon_1) - \exp(-R\epsilon_1)) \exp(r(\tau^1))\Big( \exp(r(\tau^0)) + \exp(r(\tau^1)) \Big)}{\Big( \exp(r'(\tau^0)) + \exp(r'(\tau^1)) \Big)\Big( \exp(r(\tau^0)) + \exp(r(\tau^1)) \Big)}$$
$$\leq \exp(2R\epsilon_1) - 1.$$

This implies that for any $\epsilon \leq 1$,

$$\log \mathcal{I}_{\mathcal{F}}(\epsilon/4) \leq \log \mathcal{I}_{d,B}\Big( \frac{2\ln 2}{R}\epsilon \Big) \leq \mathcal{O}\Big( d \log \frac{BR}{\epsilon} \Big),$$

where $\mathcal{I}_{d,B}(\cdot)$ is the covering number of a $d$-dimensional ball centered at the origin with radius $B$ with respect to $\ell_2$-norm and the last step is from (Wainwright, 2019). This concludes our proof.

# B  Proof of Theorem 1

The proof of Theorem 1 consists of two steps, deriving the guarantee of MLE and analyzing the performance of pessimistic offline RL.

**Step 1: MLE guarantee.** We first need to show that the confidence set $\mathcal{R}(\mathcal{D})$ contains the true reward $r^\star$ with high probability. This can be proved via the following lemma which characterizes the guarantee of MLE:

**Lemma 1** (Performance of MLE). *Fix any $\delta \in (0,1]$. Then with probability at least $1 - \delta/2$ we have that for all reward function $r \in \mathcal{G}_r$,*

$$\sum_{n=1}^{N} \log\left( \frac{P_r(o^n|\tau^{n,0}, \tau^{n,1})}{P_{r^\star}(o^n|\tau^{n,0}, \tau^{n,1})} \right) \leq c_{\mathrm{MLE}} \log(\mathcal{N}_{\mathcal{G}_r}(1/N)/\delta),$$

*where $c_{\mathrm{MLE}} > 0$ is a universal constant.*

We defer the proof to Appendix B.1. Denote the event in Lemma 1 by $\mathcal{E}_1$, then we know $\mathbb{P}(\mathcal{E}_1) \geq 1 - \delta/2$. Under the event $\mathcal{E}_1$, we have

$$\sum_{n=1}^{N} \log P_{r^\star}(o^n|\tau^{n,0}, \tau^{n,1})$$
$$\geq \sum_{n=1}^{N} \log P_{\hat{r}}(o^n|\tau^{n,0}, \tau^{n,1}) - c_{\mathrm{MLE}} \log(\mathcal{N}_{\mathcal{G}_r}(1/N)/\delta),$$

which implies that $r^\star \in \mathcal{R}(\mathcal{D})$ since we know $r^\star \in \mathcal{G}_r$ from Assumption 2.

Nevertheless, the confidence set $\mathcal{R}(\mathcal{D})$ is constructed via loglikelihood and we indeed prefer a bound on the total variation (TV) distance between $P_r$ and $P_{r^\star}$ where $r \in \mathcal{R}(\mathcal{D})$ to facilitate our subsequent analysis. We can obtain such a bound as shown in the following lemma from the literature ((Liu et al., 2022)[Proposition 14],(Zhan et al., 2022b)[Lemma 9]):

**Lemma 2.** *With probability at least $1 - \delta/2$, we have for all reward function $r \in \mathcal{G}_r$ that*

$$\mathbb{E}_{\tau^0 \sim \mu_0, \tau^1 \sim \mu_1} \left[ \left\| P_r(\cdot|\tau^0, \tau^1) - P_{r^\star}(\cdot|\tau^0, \tau^1) \right\|_1^2 \right]$$
$$\leq \frac{c_{\mathrm{TV}}}{N} \left( \sum_{n=1}^{N} \log\left( \frac{P_{r^\star}(o^n|\tau^{n,0}, \tau^{n,1})}{P_r(o^n|\tau^{n,0}, \tau^{n,1})} \right) + \log(\mathcal{N}_{\mathcal{G}_r}(1/N)/\delta) \right),$$

*where $c_{\mathrm{TV}} > 0$ is a universal constant.*

Denote the event in Lemma 2 by $\mathcal{E}_2$ and then we know $\mathbb{P}(\mathcal{E}_2) \geq 1 - \delta/2$. Then from Lemma 1 and Lemma 2 we know that under event $\mathcal{E}_1 \cap \mathcal{E}_2$, we have for all $r \in \mathcal{R}(\mathcal{D})$:

$$\mathbb{E}_{\tau^0 \sim \mu_0, \tau^1 \sim \mu_1} \left[ \left\| P_r(\cdot|\tau^0, \tau^1) - P_{r^\star}(\cdot|\tau^0, \tau^1) \right\|_1^2 \right]$$
$$\leq \frac{c \log(\mathcal{N}_{\mathcal{G}_r}(1/N)/\delta)}{N}, \tag{4}$$

where $c > 0$ is a universal constant.

Then under Assumption 3, we can apply the mean value theorem between $r^\star(\tau_1) - r^\star(\tau_0)$ and $r(\tau_1) - r(\tau_0)$ to (4) and ensure for all $r \in \mathcal{R}(\mathcal{D})$ that

$$\mathbb{E}_{\tau^0 \sim \mu_0, \tau^1 \sim \mu_1} [|(r^\star(\tau_1) - r^\star(\tau_0)) - (r(\tau_1) - r(\tau_0))|^2]$$

$$\leq \frac{c\kappa^2 \log(\mathcal{N}_{\mathcal{G}_r}(1/N)/\delta)}{N}, \tag{5}$$

where $\kappa := \frac{1}{\inf_{x \in [-r_{\max}, r_{\max}]} \Phi'(x)}$ measures the nonlinearity of the link function $\Phi$.

**Step 2: Pessimistic offline RL.** Let $r_\pi^{\inf}$ denote $\arg\min_{r \in \mathcal{R}(\mathcal{D})} J(\pi; r, P^\star) - \mathbb{E}_{\tau \sim \mu_{\mathrm{ref}}}[r(\tau)]$. Then we can bound the suboptimality of $\hat{\pi}$ as follows:

$$J(\pi_{\mathrm{tar}}; r^\star, P^\star) - J(\hat{\pi}; r^\star, P^\star)$$
$$= \left( J(\pi_{\mathrm{tar}}; r^\star, P^\star) - \mathbb{E}_{\tau \sim \mu_{\mathrm{ref}}}[r^\star(\tau)] \right)$$
$$\quad - \left( J(\hat{\pi}; r^\star, P^\star) - \mathbb{E}_{\tau \sim \mu_{\mathrm{ref}}}[r^\star(\tau)] \right)$$
$$\leq \left( \left( J(\pi_{\mathrm{tar}}; r^\star, P^\star) - \mathbb{E}_{\tau \sim \mu_{\mathrm{ref}}}[r^\star(\tau)] \right) \right.$$
$$\quad \left. - \left( J(\pi_{\mathrm{tar}}; r_{\pi_{\mathrm{tar}}}^{\inf}, P^\star) - \mathbb{E}_{\tau \sim \mu_{\mathrm{ref}}}[r_{\pi_{\mathrm{tar}}}^{\inf}(\tau)] \right) \right)$$
$$\quad - \left( \left( J(\hat{\pi}; r^\star, P^\star) - \mathbb{E}_{\tau \sim \mu_{\mathrm{ref}}}[r^\star(\tau)] \right) \right.$$
$$\quad \left. - \left( J(\hat{\pi}; r_{\hat{\pi}}^{\inf}, P^\star) - \mathbb{E}_{\tau \sim \mu_{\mathrm{ref}}}[r_{\hat{\pi}}^{\inf}(\tau)] \right) \right)$$
$$\leq \left( J(\pi_{\mathrm{tar}}; r^\star, P^\star) - \mathbb{E}_{\tau \sim \mu_{\mathrm{ref}}}[r^\star(\tau)] \right)$$
$$\quad - \left( J(\pi_{\mathrm{tar}}; r_{\pi_{\mathrm{tar}}}^{\inf}, P^\star) - \mathbb{E}_{\tau \sim \mu_{\mathrm{ref}}}[r_{\pi_{\mathrm{tar}}}^{\inf}(\tau)] \right)$$
$$= \mathbb{E}_{\tau^0 \sim \pi_{\mathrm{tar}}, \tau^1 \sim \mu_{\mathrm{ref}}} [(r^\star(\tau^0) - r^\star(\tau^1))$$
$$\quad - (r_{\pi_{\mathrm{tar}}}^{\inf}(\tau^0) - r_{\pi_{\mathrm{tar}}}^{\inf}(\tau^1))]$$
$$\leq C_r(\mathcal{G}_r, \pi_{\mathrm{tar}}, \mu_{\mathrm{ref}})$$
$$\quad \cdot \sqrt{\mathbb{E}_{\tau_0 \sim \mu_0, \tau_1 \sim \mu_1} [|r^\star(\tau^0) - r^\star(\tau^1) - r_{\pi_{\mathrm{tar}}}^{\inf}(\tau^0) + r_{\pi_{\mathrm{tar}}}^{\inf}(\tau^1)|^2]}$$
$$\leq \sqrt{\frac{c C_r^2(\mathcal{G}_r, \pi_{\mathrm{tar}}, \mu_{\mathrm{ref}}) \kappa^2 \log(\mathcal{N}_{\mathcal{G}_r}(1/N)/\delta)}{N}},$$

where the second step is due to $\hat{\pi} = \arg\max_{\pi \in \Pi_{\mathrm{his}}} \min_{r \in \mathcal{R}(\mathcal{D})} J(\pi; r, P^\star) - \mathbb{E}_{\tau \sim \mu_{\mathrm{ref}}}[r(\tau)]$, the third step is due to $r_{\hat{\pi}}^{\inf} = \arg\min_{r \in \mathcal{R}(\mathcal{D})} J(\hat{\pi}; r, P^\star) - \mathbb{E}_{\tau \sim \mu_{\mathrm{ref}}}[r(\tau)]$, the fifth step comes from the definition of $C_r(\mathcal{G}_r, \pi_{\mathrm{tar}}, \mu_{\mathrm{ref}})$ (Definition 2) and the last step leverages (5). This concludes our proof.

### B.1 Proof of Lemma 1

The proof largely follows (Zhan et al., 2022b). Suppose $\overline{\mathcal{F}}$ is a $1/N$-bracket of $\mathcal{G}_r$ with $|\overline{\mathcal{F}}| = \mathcal{N}_{\mathcal{G}_r}(1/N)$ and we denote the set of all right brackets in $\overline{\mathcal{F}}$ by $\widetilde{\mathcal{F}}$, i.e., $\widetilde{\mathcal{F}} := \{ f : \exists f', \text{ such that } [f', f] \in \overline{\mathcal{F}} \}$. Then fix any $f \in \widetilde{\mathcal{F}}$, we have:

$$\mathbb{E}\left[ \exp\left( \sum_{n=1}^{N} \log\left( \frac{f(o^n|\tau^{n,0}, \tau^{n,1})}{P_{r^\star}(o^n|\tau^{n,0}, \tau^{n,1})} \right) \right) \right]$$
$$= \prod_{n=1}^{N} \mathbb{E}\left[ \exp\left( \log\left( \frac{f(o^n|\tau^{n,0}, \tau^{n,1})}{P_{r^\star}(o^n|\tau^{n,0}, \tau^{n,1})} \right) \right) \right]$$
$$= \prod_{n=1}^{N} \mathbb{E}\left[ \frac{f(o^n|\tau^{n,0}, \tau^{n,1})}{P_{r^\star}(o^n|\tau^{n,0}, \tau^{n,1})} \right]$$

$$= \prod_{n=1}^{N} \mathbb{E}\left[\sum_o f(o|\tau^{n,0},\tau^{n,1})\right] \le \left(1 + \frac{1}{N}\right)^N \le e,$$

where the first step is due to each sample in $\mathcal{D}$ is i.i.d., the third step uses Tower property and the fourth step is from the fact that $\overline{\mathcal{F}}$ is a minimum $1/N$-bracket.

Then by Markov's inequality we have for any $\delta \in (0,1]$,

$$\mathbb{P}\left(\sum_{n=1}^{N}\log\left(\frac{f(o^n|\tau^{n,0},\tau^{n,1})}{P_{r^\star}(o^n|\tau^{n,0},\tau^{n,1})}\right) > \log(1/\delta)\right)$$
$$\le \mathbb{E}\left[\exp\left(\sum_{n=1}^{N}\log\left(\frac{f(o^n|\tau^{n,0},\tau^{n,1})}{P_{r^\star}(o^n|\tau^{n,0},\tau^{n,1})}\right)\right)\right]$$
$$\cdot \exp[-\log(1/\delta)]$$
$$\le e\delta.$$

By union bound, we have for all $f \in \widetilde{\mathcal{F}}$,

$$\mathbb{P}\left(\sum_{n=1}^{N}\log\left(\frac{f(o^n|\tau^{n,0},\tau^{n,1})}{P_{r^\star}(o^n|\tau^{n,0},\tau^{n,1})}\right)\right.$$
$$\left. > c_{\text{MLE}}\log(\mathcal{N}_{\mathcal{G}_r}(1/N)/\delta)\right) \le \delta/2,$$

where $c_{\text{MLE}} > 0$ is a universal constant.

Therefore from the definition of $1/N$-bracket net, we know for all $r \in \mathcal{G}_r$, there exists $f \in \widetilde{\mathcal{F}}$ such that $P_r(\cdot|\tau^0,\tau^1) \le f(\cdot|\tau^0,\tau^1)$ for any trajectories $(\tau^0,\tau^1)$. This implies that for all $r \in \mathcal{G}_r$,

$$\mathbb{P}\left(\sum_{n=1}^{N}\log\left(\frac{P_r(o^n|\tau^{n,0},\tau^{n,1})}{P_{r^\star}(o^n|\tau^{n,0},\tau^{n,1})}\right)\right.$$
$$\left. > c_{\text{MLE}}\log(\mathcal{N}_{\mathcal{G}_r}(1/N)/\delta)\right) \le \delta/2.$$

This concludes our proof.

## C Comparison with (Zhu et al., 2023)

(Zhu et al., 2023) considers the linear reward setting under BTL model and they can achieve the following sample complexity:

$$N = \mathcal{O}\left(\frac{C_{\text{lin}}^2 \exp(4BR)d\log(1/\delta)}{\epsilon^2}\right),$$

where $R$ and $B$ are the norm bounds on the feature vectors $\phi$ and parameter $\theta$ (defined in Proposition 1). The concentrability coefficient $C_{\text{lin}}$ is defined as

$$C_{\text{lin}} := \|\mathbb{E}_{\tau^0\sim\pi_{\text{tar}},\tau^1\sim\mu_{\text{ref}}}[\phi(\tau^0) - \phi(\tau^1)]\|_{\Sigma_{\mathcal{D}}^{-1}},$$

and $\Sigma_{\mathcal{D}}$ is the empirical covariance matrix of the dataset $\frac{1}{N}\sum_{n=1}^{N}(\phi(\tau^{n,0}) - \phi(\tau^{n,1}))(\phi(\tau^{n,0}) - \phi(\tau^{n,1}))^\top$.

Note that all the analysis and proofs in this paper still hold when we define the concentrability coefficient as

$$C_r'(\mathcal{G}_r,\pi_{\text{tar}},\mu_{\text{ref}}) := \max\left\{0,\right.$$
$$\left.\sup_{r\in\mathcal{G}_r}\frac{\mathbb{E}_{\tau^0\sim\pi_{\text{tar}},\tau^1\sim\mu_{\text{ref}}}[r^\star(\tau^0) - r^\star(\tau^1) - r(\tau^0) + r(\tau^1)]}{\sqrt{\frac{1}{N}\sum_{n=1}^{N}|r^\star(\tau^{n,0}) - r^\star(\tau^{n,1}) - r(\tau^{n,0}) + r(\tau^{n,1})|^2}}\right\}.$$

Then when specializing the result in Theorem 1 to the linear reward setting under BTL model with this version of concentrability coefficient, the sample complexity is

$$N = \widetilde{\mathcal{O}}\left(\frac{(C_r'(\mathcal{G}_r,\pi_{\text{tar}},\mu_{\text{ref}}))^2 \exp(2r_{\max})d\log(BR/\delta)}{\epsilon^2}\right).$$

We know that $BR \ge r_{\max}$. In addition, note that in this case, we have $C_{\text{lin}} \ge 0$ and for any $r \in \mathcal{G}_r$,

$$\left|\mathbb{E}_{\tau^0\sim\pi_{\text{tar}},\tau^1\sim\mu_{\text{ref}}}[r^\star(\tau^0) - r^\star(\tau^1) - r(\tau^0) + r(\tau^1)]\right|$$
$$= \left|\langle\mathbb{E}_{\tau^0\sim\pi_{\text{tar}},\tau^1\sim\mu_{\text{ref}}}[\phi(\tau^0) - \phi(\tau^1)],\theta^\star - \theta\rangle\right|$$
$$\le \|\mathbb{E}_{\tau^0\sim\pi_{\text{tar}},\tau^1\sim\mu_{\text{ref}}}[\phi(\tau^0) - \phi(\tau^1)]\|_{\Sigma_{\mathcal{D}}^{-1}} \cdot \|\theta^\star - \theta\|_{\Sigma_{\mathcal{D}}}$$
$$= \|\mathbb{E}_{\tau^0\sim\pi_{\text{tar}},\tau^1\sim\mu_{\text{ref}}}[\phi(\tau^0) - \phi(\tau^1)]\|_{\Sigma_{\mathcal{D}}^{-1}}$$
$$\cdot \sqrt{\frac{1}{N}\sum_{n=1}^{N}|r^\star(\tau^{n,0}) - r^\star(\tau^{n,1}) - r(\tau^{n,0}) + r(\tau^{n,1})|^2},$$

where we suppose $r^\star(\tau) = \langle\phi(\tau),\theta^\star\rangle$ and $r(\tau) = \langle\phi(\tau),\theta\rangle$. Therefore we have

$$C_r'(\mathcal{G}_r,\pi_{\text{tar}},\mu_{\text{ref}}) \le C_{\text{lin}}.$$

This implies that Theorem 1 can recover the sample complexity for linear reward setting under BTL model in (Zhu et al., 2023) with only some additional log factors.

## D Omitted Details

In this section we supplement the definition of bracket number for the transition class and advantage function class.

**Definition 5** ($\epsilon$-bracket number of transition probability classes). *Suppose $f^1, f^2$ is a function with $f^1(\cdot|s,a), f^2(\cdot|s,a) \in \mathbb{R}^{|\mathcal{S}|}$ for all $(s,a) \in \mathcal{S} \times \mathcal{A}$. Then we say $(f^1, f^2)$ is a $\epsilon$-bracket if $f^1(\cdot|s,a) \le f^2(\cdot|s,a)$ and $\|f^1(\cdot|s,a) - f^2(\cdot|s,a)\|_1 \le \epsilon$ for all $(s,a)$. The $\epsilon$-bracket number of a transition probability class $\mathcal{G}_{P_h}$ where $h \in [H-1]$ is the minimum integer $N$ satisfying that there exist $N$ $\epsilon$-brackets $(f^{n,1}, f^{n,2})_{n=1}^{N}$ such that for any function $P_h \in \mathcal{G}_{P_h}$ there is a bracket $(f^{i,1}, f^{i,2})$ where $i \in [N]$ containing it, i.e., $f^{i,1}(\cdot|s,a) \le P_h(\cdot|s,a) \le f^{i,2}(\cdot|s,a)$ for all $(s,a)$.*

**Definition 6** ($\epsilon$-bracket number of initial state distribution classes). *Suppose $f^1, f^2 \in \mathbb{R}^{|\mathcal{S}|}$. Then we say $(f^1, f^2)$ is a $\epsilon$-bracket if $f^1 \le f^2$ and $\|f^1 - f^2\|_1 \le \epsilon$. The*

$\epsilon$-*bracket number of a initial state distribution class* $\mathcal{G}_{P_0}$ *is the minimum integer* $N$ *satisfying that there exist* $N$ $\epsilon$-*brackets* $(f^{n,1}, f^{n,2})_{n=1}^N$ *such that for any* $P_0 \in \mathcal{G}_{P_0}$ *there is a bracket* $(f^{i,1}, f^{i,2})$ *where* $i \in [N]$ *containing it, i.e.,* $f^{i,1} \leq P_0 \leq f^{i,2}$.

**Definition 7** ($\epsilon$-bracket number of advantage function classes)**.** *Suppose* $g^1, g^2$ *is a function with* $g^1(\cdot|s, a^0, a^1), g^2(\cdot|s, a^0, a^1) \in \mathbb{R}^2$ *for all* $(s, a^0, a^1) \in \mathcal{S} \times \mathcal{A} \times \mathcal{A}$. *Then we say* $(g^1, g^2)$ *is a* $\epsilon$-*bracket if* $g^1(\cdot|s, a^0, a^1) \leq g^2(\cdot|s, a^0, a^1)$ *and* $\|g^1(\cdot|s, a^0, a^1) - g^2(\cdot|s, a^0, a^1)\|_1 \leq \epsilon$ *for all* $(s, a^0, a^1) \in \mathcal{S} \times \mathcal{A} \times \mathcal{A}$. *The* $\epsilon$-*bracket number of a reward class* $\mathcal{G}_{A_h}$ *where* $h \in [H]$ *is the minimum integer* $N$ *satisfying that there exist* $N$ $\epsilon$-*brackets* $(g^{n,1}, g^{n,2})_{n=1}^N$ *such that for any function* $A_h \in \mathcal{G}_{A_h}$ *there is a bracket* $(g^{i,1}, g^{i,2})$ *where* $i \in [N]$ *containing it, i.e.,* $g^{i,1}(\cdot|s, a^0, a^1) \leq P_{A_h}(\cdot|s, a^0, a^1) \leq g^{i,2}(\cdot|s, a^0, a^1)$ *for all* $(s, a^0, a^1) \in \mathcal{S} \times \mathcal{A} \times \mathcal{A}$.

We use $\mathcal{N}_{\mathcal{G}_{P_h}}(\epsilon)$ and $\mathcal{N}_{\mathcal{G}_{A_h}}(\epsilon)$ to denote the $\epsilon$-bracket number of $\mathcal{G}_{P_h}$ and $\mathcal{G}_{A_h}$. Similarly, when the transition probability or the advantage function possesses a low-dimension embedding, we can also bound the $\epsilon$-bracket number efficiently.

# E Proofs of Lower Bounds

## E.1 Proof of Proposition 2

Given any $S, A, H$, consider a Markov Chain with horizon $H$, state space $\mathcal{S}$ and transition $P_{MC}$ where $|\mathcal{S}| = S$ and $P_{MC,h} : \mathcal{S} \times \mathcal{S} \mapsto [0, 1]$ specifies the transition probability for each step $h$. Then we can construct a MDP which has the same state space $\mathcal{S}$ and horizon $H$. In addition, the action space $\mathcal{A}$ satisfies $|\mathcal{A}| = A$ and the transition $P^\star$ is independent from the action, i.e., $P_h^\star(s'|s, a) = P_{MC,h}(s'|s)$ for all $a \in \mathcal{A}$.

We consider the case where $\pi_{tar}$ is a Markovian policy. Then we can define $\pi_{b,h}(\cdot|s)$ for every $h \in [H], s \in \mathcal{S}$ as follows:

$$\pi_{b,h}(a|s) = \begin{cases} \pi_{tar,h}(a|s)/C, & \text{if } a = a^1, \\ \pi_{tar,h}(a|s) + (1 - 1/C)\pi_{tar,h}(a^1|s), & \text{if } a = a^2, \\ \pi_{tar,h}(a|s), & \text{otherwise}, \end{cases}$$

and we select $\mu_0 = d^{\{\pi_{b,h}\}_{h=1}^H}$.

Then since $d_h^\pi(s)$ is fixed for any policy $\pi$, we know

$$C_{st} = \sup_{s,a,h} \frac{\pi_{tar,h}(a|s)}{\pi_{b,h}(a|s)} = C.$$

On the other hand, we have

$$C_{tr} = \sup_{a_{1:H}, s_{1:H}} \prod_{h \in [H]} \frac{\pi_{tar,h}(a_h|s_h)}{\pi_{b,h}(a_h|s_h)}$$

$$= \sup_{s_{1:H}} \prod_{h \in [H]} \frac{\pi_{tar,h}(a^1|s_h)}{\pi_{b,h}(a^1|s_h)} = C^H.$$

This concludes our proof.

## E.2 Proof of Theorem 2

The proof is inspired by the hard instances in (Rashidinejad et al., 2021b). We consider the case $C \geq 2$ and $1 < C < 2$ respectively.

**Case 1:** $C \geq 2$. Consider the case where there is only one state $s$ and two actions $a^1, a^2$. Set the dataset distribution $\mu_0 = \mu_1$ where

$$\mu_{0,h}(s, a^1) = \frac{1}{C}, \qquad \mu_{0,h}(s, a^2) = 1 - \frac{1}{C}, \qquad \forall h \in [H],$$

and $\mu_0(\tau) = \prod_{h=1}^H \mu_{0,h}(s_h, a_h)$ for all trajectory $\tau = (s_1, a_1, \cdots, s_H, a_H)$.

We consider two different reward function $r^1$ and $r^2$:

$$r^1(\tau) = \begin{cases} \frac{1}{2} + x, & \text{if all the actions in } \tau \text{ are } a^1, \\ \frac{1}{2}, & \text{otherwise}. \end{cases}$$

$$r^2(\tau) = \begin{cases} \frac{1}{2} - x, & \text{if all the actions in } \tau \text{ are } a^1, \\ \frac{1}{2}, & \text{otherwise}. \end{cases}$$

Here $0 < x < \frac{1}{2}$ is a quantity we will specify later and we denote the special trajectory where all the actions are $a^1$ by $\tau^\star$. Then we have two MDPs, $\mathcal{M}_1$ and $\mathcal{M}_2$ whose reward functions are $r^1$ and $r^2$ respectively. It can be easily verified that $(\mathcal{M}_1, \mu_0) \in \Theta_{st}(C)$, $(\mathcal{M}_2, \mu_0) \in \Theta_{st}(C)$.

Further, let $L(\pi; \mathcal{M})$ denote the suboptimality of policy $\pi$ in $\mathcal{M}$, then we have for all policies $\pi$,

$$L(\pi; \mathcal{M}_1) + L(\pi; \mathcal{M}_2) \geq x.$$

Now we can apply Fano's inequality, which leads to the following inequality

$$\inf_{\hat{\pi}} \sup_{\mathcal{M} \in \{\mathcal{M}_1, \mathcal{M}_2\}} \mathbb{E}_{\mathcal{D}}[L(\pi, \mathcal{M})]$$

$$\geq \frac{x}{2}\left(2 - N \cdot \mathbf{KL}\left(\mu_0 \otimes \mu_1 \otimes P_{r^1} \| \mu_0 \otimes \mu_1 \otimes P_{r^2}\right)\right).$$

Now we only need to bound $\mathbf{KL}\left(\mu_0 \otimes \mu_1 \otimes P_{r^1} \| \mu_0 \otimes \mu_1 \otimes P_{r^2}\right)$, which can be computed as follows:

$$\mathbf{KL}\left(\mu_0 \otimes \mu_1 \otimes P_{r^1} \| \mu_0 \otimes \mu_1 \otimes P_{r^2}\right)$$

$$= 2 \sum_{\tau^0 = \tau^\star, \tau^1 \neq \tau^\star} \mu_0(\tau^0)\mu_1(\tau^1)\mathbf{KL}\left(\text{Bern}(\sigma(x)) \| \text{Bern}(\sigma(-x))\right)$$

$$\leq \frac{2\exp(1/2)x^2}{C^H}.$$

Then by letting $x = \min\left\{\frac{1}{2}, \sqrt{\frac{C^H}{2\exp(1/2)N}}\right\}$, we have

$$\inf_{\widehat{\pi}} \sup_{\mathcal{M}\in\{\mathcal{M}_1,\mathcal{M}_2\}} \mathbb{E}_{\mathcal{D}}[L(\pi,\mathcal{M})]$$

$$\geq \frac{x}{2} = \min\left\{\frac{1}{4}, \sqrt{\frac{C^H}{8\exp(1/2)N}}\right\}.$$

**Case 2:** $1 < C < 2$**.** Consider the case where there are two one states $s^1, s^2$ and two actions $a^1, a^2$. We suppose the initial state distribution of $P_0^\star$ is fixed as $P_0^\star(s^1) = C-1$ and $P_0^\star(s^2) = 2-C$. In addition, the state will stay the same throughout the whole episode. Then we can set the dataset distribution $\mu_0 = \mu_1$ where

$$\mu_0(\tau) = \begin{cases} \frac{2(C-1)}{C}\cdot\frac{1}{2^H}, & \text{if the state of } \tau \text{ is } s^1, \\ \frac{2-C}{C}, & \text{if the state of } \tau \text{ is } s^2 \\ & \text{and the actions are all } a^1, \\ 0, & \text{if the state of } \tau \text{ is } s^2 \\ & \text{and the actions contain } a^2. \end{cases}$$

Then we know

$$\mu_{0,h}(s^1,a^1) = \mu_{0,h}(s^1,a^2) = \frac{C-1}{C},$$

$$\mu_{0,h}(s^2,a^1) = \frac{2-C}{C}, \mu_{0,h}(s^2,a^2) = 0, \qquad \forall h \in [H].$$

We consider two different reward function $r^1$ and $r^2$:

$$r^1(\tau) = \begin{cases} \frac{1}{2}+x, & \text{if the state is } s^1 \text{ and} \\ & \text{all the actions in } \tau \text{ are } a^1, \\ \frac{1}{2}, & \text{otherwise.} \end{cases}$$

$$r^2(\tau) = \begin{cases} \frac{1}{2}-x, & \text{if the state is } s^1 \text{ and} \\ & \text{all the actions in } \tau \text{ are } a^1, \\ \frac{1}{2}, & \text{otherwise.} \end{cases}$$

Here $0 < x < \frac{1}{2}$ is a quantity we will specify later and we denote the special trajectory where the state is $s^1$ and all the actions are $a^1$ by $\tau^\star$. Then we have two MDPs, $\mathcal{M}_1$ and $\mathcal{M}_2$ whose reward functions are $r^1$ and $r^2$ respectively. It can be easily verified that $(\mathcal{M}_1,\mu_0) \in \Theta_{\mathrm{st}}(C), (\mathcal{M}_2,\mu_0) \in \Theta_{\mathrm{st}}(C)$.

In addition, we have for all policies $\pi$,

$$L(\pi;\mathcal{M}_1) + L(\pi;\mathcal{M}_2) \geq (C-1)x.$$

Therefore by Fano's inequality, we have

$$\inf_{\widehat{\pi}} \sup_{\mathcal{M}\in\{\mathcal{M}_1,\mathcal{M}_2\}} \mathbb{E}_{\mathcal{D}}[L(\pi,\mathcal{M})]$$

$$\geq \frac{(C-1)x}{2}\left(2 - N\cdot\mathbf{KL}\Big(\mu_0\otimes\mu_1\otimes P_{r^1}\,\|\,\mu_0\otimes\mu_1\otimes P_{r^2}\Big)\right),$$

where the KL divergence can be computed as follows:

$$\mathbf{KL}\Big(\mu_0\otimes\mu_1\otimes P_{r^1}\,\|\,\mu_0\otimes\mu_1\otimes P_{r^2}\Big)$$

$$=2\sum_{\tau^0=\tau^\star,\tau^1\neq\tau^\star}\mu_0(\tau^0)\mu_1(\tau^1)\mathbf{KL}\big(\mathrm{Bern}(\sigma(x))\|\mathrm{Bern}(\sigma(-x)))$$

$$\leq\frac{4(C-1)\exp(1/2)x^2}{2^H C}.$$

Then by letting $x = \min\left\{\frac{1}{2}, \sqrt{\frac{2^H C}{4\exp(1/2)(C-1)N}}\right\}$, we have

$$\inf_{\widehat{\pi}} \sup_{\mathcal{M}\in\{\mathcal{M}_1,\mathcal{M}_2\}} \mathbb{E}_{\mathcal{D}}[L(\pi,\mathcal{M})]$$

$$\geq \frac{(C-1)x}{2} = \min\left\{\frac{C-1}{4}, \sqrt{\frac{2^H C(C-1)}{16\exp(1/2)N}}\right\}.$$

In conclusion, we have for any $C > 1$ and $H \geq 1$,

$$\inf_{\widehat{\pi}} \sup_{(\mathcal{M},\mu_0)\in\Theta_{\mathrm{st}}(C)} \mathbb{E}_{\mathcal{D}}[J(\pi^\star;r^\star,P^\star) - J(\widehat{\pi};r^\star,P^\star)]$$

$$\gtrsim \min\left\{C-1, \sqrt{\frac{(\max\{C,2\})^{H-1}(C-1)}{N}}\right\}.$$

### E.3 Proof of Theorem 3

The proof is quite similar to the proof of Theorem 2. We consider the case $C \geq 2$ and $1 < C < 2$ respectively.

**Case 1:** $C \geq 2$**.** Consider the case where there is only one state $s$ and two actions $a^1, a^2$. Set the dataset distribution $\mu_0 = \mu_1$ where

$$\mu_0(\tau^\star) = \frac{1}{C}, \qquad \mu_0(\tau^\dagger) = 1 - \frac{1}{C},$$

where $\tau^\star$ is the trajecotry where the actions are all $a^1$ and $\tau^\dagger$ is the trajecotry where the actions are all $a^2$.

We consider two different reward function $r^1$ and $r^2$:

$$r^1(\tau) = \begin{cases} \frac{1}{2}+x, & \text{if all the actions in } \tau \text{ are } a^1, \\ \frac{1}{2}, & \text{otherwise.} \end{cases}$$

$$r^2(\tau) = \begin{cases} \frac{1}{2}-x, & \text{if all the actions in } \tau \text{ are } a^1, \\ \frac{1}{2}, & \text{otherwise.} \end{cases}$$

Here $0 < x < \frac{1}{2}$ is a quantity we will specify later. Then we have two MDPs, $\mathcal{M}_1$ and $\mathcal{M}_2$ whose reward functions are $r^1$ and $r^2$ respectively. It can be easily verified that $(\mathcal{M}_1,\mu_0) \in \Theta_{\mathrm{tr}}(C), (\mathcal{M}_2,\mu_0) \in \Theta_{\mathrm{tr}}(C)$.

Further, let $L(\pi; \mathcal{M})$ denote the suboptimality of policy $\pi$ in $\mathcal{M}$, then we have for all policies $\pi$,

$$L(\pi; \mathcal{M}_1) + L(\pi; \mathcal{M}_2) \geq x.$$

Now we can apple Fano's inequality, which leads to the following inequality

$$\inf_{\widehat{\pi}} \sup_{\mathcal{M} \in \{\mathcal{M}_1, \mathcal{M}_2\}} \mathbb{E}_{\mathcal{D}}[L(\pi, \mathcal{M})]$$
$$\geq \frac{x}{2}\left(2 - N \cdot \mathbf{KL}\left(\mu_0 \otimes \mu_1 \otimes P_{r^1} \| \mu_0 \otimes \mu_1 \otimes P_{r^2}\right)\right).$$

Now we only need to bound $\mathbf{KL}\left(\mu_0 \otimes \mu_1 \otimes P_{r^1} \| \mu_0 \otimes \mu_1 \otimes P_{r^2}\right)$, which can be computed as follows:

$$\mathbf{KL}\left(\mu_0 \otimes \mu_1 \otimes P_{r^1} \| \mu_0 \otimes \mu_1 \otimes P_{r^2}\right)$$
$$=2 \sum_{\tau^0=\tau^\star, \tau^1=\tau^\dagger} \mu_0(\tau^0)\mu_1(\tau^1)$$
$$\cdot \mathbf{KL}\left(\text{Bern}(\sigma(x)) \| \text{Bern}(\sigma(-x))\right)$$
$$\leq \frac{2\exp(1/2)x^2}{C}.$$

Then by letting $x = \min\left\{\frac{1}{2}, \sqrt{\frac{C}{2\exp(1/2)N}}\right\}$, we have

$$\inf_{\widehat{\pi}} \sup_{\mathcal{M} \in \{\mathcal{M}_1, \mathcal{M}_2\}} \mathbb{E}_{\mathcal{D}}[L(\pi, \mathcal{M})]$$
$$\geq \frac{x}{2} = \min\left\{\frac{1}{4}, \sqrt{\frac{C}{8\exp(1/2)N}}\right\}.$$

**Case 2:** $1 < C < 2$. Consider the case where there are two one states $s^1, s^2$ and two actions $a^1, a^2$. We suppose the initial state distribution of $P_0^\star$ is fixed as $P_0^\star(s^1) = C - 1$ and $P_0^\star(s^2) = 2 - C$. In addition, the state will stay the same throughout the whole episode. Then we can set the dataset distribution $\mu_0 = \mu_1$ where

$$\mu_0(\tau) = \begin{cases} \frac{2(C-1)}{C} \cdot \frac{1}{2}, & \text{if the state of } \tau \text{ is } s^1 \\ & \text{and the actions are all } a^1 \text{ or all } a^2, \\ \frac{2-C}{C}, & \text{if the state of } \tau \text{ is } s^2 \\ & \text{and the actions are all } a^1, \\ 0, & \text{if the state of } \tau \text{ is } s^2 \\ & \text{and the actions contain } a^2. \end{cases}$$

Let $\tau^\star$ be the trajectory where the state is $s^1$ and the actions are all $a^1$.

We further consider two different reward function $r^1$ and $r^2$:

$$r^1(\tau) = \begin{cases} \frac{1}{2} + x, & \text{if the state is } s^1 \text{ and} \\ & \text{all the actions in } \tau \text{ are } a^1, \\ \frac{1}{2}, & \text{otherwise.} \end{cases}$$

$$r^2(\tau) = \begin{cases} \frac{1}{2} - x, & \text{if the state is } s^1 \text{ and} \\ & \text{all the actions in } \tau \text{ are } a^1, \\ \frac{1}{2}, & \text{otherwise.} \end{cases}$$

Here $0 < x < \frac{1}{2}$ is a quantity we will specify later. Then we have two MDPs, $\mathcal{M}_1$ and $\mathcal{M}_2$ whose reward functions are $r^1$ and $r^2$ respectively. It can be easily verified that $(\mathcal{M}_1, \mu_0) \in \Theta_{\text{tr}}(C), (\mathcal{M}_2, \mu_0) \in \Theta_{\text{tr}}(C)$.

In addition, we have for all policies $\pi$,

$$L(\pi; \mathcal{M}_1) + L(\pi; \mathcal{M}_2) \geq (C - 1)x.$$

Therefore by Fano's inequality, we have

$$\inf_{\widehat{\pi}} \sup_{\mathcal{M} \in \{\mathcal{M}_1, \mathcal{M}_2\}} \mathbb{E}_{\mathcal{D}}[L(\pi, \mathcal{M})]$$
$$\geq \frac{(C-1)x}{2}\left(2 - N \cdot \mathbf{KL}\left(\mu_0 \otimes \mu_1 \otimes P_{r^1} \| \mu_0 \otimes \mu_1 \otimes P_{r^2}\right)\right),$$

where the KL divergence can be computed as follows:

$$\mathbf{KL}\left(\mu_0 \otimes \mu_1 \otimes P_{r^1} \| \mu_0 \otimes \mu_1 \otimes P_{r^2}\right)$$
$$=2 \sum_{\tau^0=\tau^\star, \tau^1 \neq \tau^\star} \mu_0(\tau^0)\mu_1(\tau^1)\mathbf{KL}\left(\text{Bern}(\sigma(x)) \| \text{Bern}(\sigma(-x))\right)$$
$$\leq \frac{2(C-1)\exp(1/2)x^2}{C}.$$

Then by letting $x = \min\left\{\frac{1}{2}, \sqrt{\frac{C}{2\exp(1/2)(C-1)N}}\right\}$, we have

$$\inf_{\widehat{\pi}} \sup_{\mathcal{M} \in \{\mathcal{M}_1, \mathcal{M}_2\}} \mathbb{E}_{\mathcal{D}}[L(\pi, \mathcal{M})]$$
$$\geq \frac{(C-1)x}{2} = \min\left\{\frac{C-1}{4}, \sqrt{\frac{(C-1)}{8\exp(1/2)N}}\right\}.$$

In conclusion, we have for any $C > 1$ and $H \geq 1$,

$$\inf_{\widehat{\pi}} \sup_{(\mathcal{M}, \mu_0) \in \Theta_{\text{st}}(C)} \mathbb{E}_{\mathcal{D}}[J(\pi^\star; r^\star, P^\star) - J(\widehat{\pi}; r^\star, P^\star)]$$
$$\gtrsim \min\left\{C - 1, \sqrt{\frac{C-1}{N}}\right\}.$$

# F Proof of Theorem 4

The proof still consists of two steps, deriving the guarantee of MLE and analyzing the performance of pessimistic offline RL.

**Step 1: MLE guarantee.** Note that Lemma 1 and Lemma 2 still applies here. Let $\mathcal{E}_1$ and $\mathcal{E}_2$ denote the event in Lemma 1 and Lemma 2 respectively. Following almost the same arguments, we have the following guarantee for the estimation of the system dynamics:

**Lemma 3.** *Under Assumption 4, with probability at least* $1 - \delta/2$*, the following event holds true:*

$$(1) P_h^\star \in \mathcal{P}_h(\mathcal{D}), P_0^\star \in \mathcal{P}_{\text{ini}}(\mathcal{D}), \qquad \forall h \in [H-1],$$

$$(2) \mathbb{E}_{(s_h, a_h) \sim \mu_{0,h}} \left[ \left\| P_h(\cdot|s,a) - P_h^\star(\cdot|s,a) \right\|_1^2 \right]$$

$$+ \mathbb{E}_{(s_h, a_h) \sim \mu_{1,h}} \left[ \left\| P_h(\cdot|s,a) - P_h^\star(\cdot|s,a) \right\|_1^2 \right]$$

$$\leq \frac{c \log(H \mathcal{N}_{\mathcal{G}_{P_h}}(1/N)/\delta)}{N}, \forall h \in [H-1], P_h \in \mathcal{P}_h(\mathcal{D}),$$

$$(3) \mathbb{E}_{s \sim \mu_{0,1}} \left[ \left\| P_0(s) - P_0^\star(s) \right\|_1^2 \right]$$

$$+ \mathbb{E}_{s \sim \mu_{1,1}} \left[ \left\| P_0(s) - P_0^\star(s) \right\|_1^2 \right]$$

$$\leq \frac{c \log(H \mathcal{N}_{\mathcal{G}_{P_0}}(1/N)/\delta)}{N}, \forall P_0 \in \mathcal{P}_0(\mathcal{D}).$$

The proof is omitted here. Let $\mathcal{E}_3$ denote the event in Lemma 3.

**Step 2: Pessimistic offline RL.** We first introduce the following lemma which suggests that under event $\mathcal{E}_3$, we can evaluate the expected cumulative reward of $\pi_{\text{tar}}$ with respect to any reward function $r \in \mathcal{G}_r$ via the system dynamics $P_h \in \mathcal{P}_h(\mathcal{D})$:

**Lemma 4.** *Suppose Asusmption 3 is true. Then under* $\mathcal{E}_3$*, we have for all reward function* $r \in \mathcal{G}_r$ *and* $P = (\{P_h\}_{h=0}^{H-1})$ *where* $P_h \in \mathcal{P}_h(\mathcal{D})$ *that*

$$J(\pi_{\text{tar}}; r, P^\star) - J(\pi_{\text{tar}}; r, P)$$

$$\leq H r_{\max} \sqrt{\frac{c C_P^2(\{\mathcal{G}_{P_h}\}, \pi_{\text{tar}}) \log(H \mathcal{N}_P(1/N)/\delta)}{N}},$$

*where* $\mathcal{N}_P = \max_{0 \leq h \leq H-1}\{\mathcal{N}_{\mathcal{G}_{P_h}}\}$*.*

The proof is deferred to Appendix F.1.

Let $(r_\pi^{\inf}, P_\pi^{\inf})$ denote $\text{argmin}_{r \in \mathcal{R}(\mathcal{D}), P \in \mathcal{P}_{\text{ini}}(\mathcal{D}) \times \prod_{h=1}^{H-1} \mathcal{P}_h(\mathcal{D})} J(\pi; r, P) - \mathbb{E}_{\tau \sim \mu_{\text{ref}}}[r(\tau)]$. Then under the event $\mathcal{E}_3$, we can bound the suboptimality of $\widehat{\pi}$ as follows:

$$J(\pi_{\text{tar}}; r^\star, P^\star) - J(\widehat{\pi}; r^\star, P^\star)$$

$$= \left( J(\pi_{\text{tar}}; r^\star, P^\star) - \mathbb{E}_{\tau \sim \mu_{\text{ref}}}[r^\star(\tau)] \right)$$

$$- \left( J(\widehat{\pi}; r^\star, P^\star) - \mathbb{E}_{\tau \sim \mu_{\text{ref}}}[r^\star(\tau)] \right)$$

$$= \left( \left( J(\pi_{\text{tar}}; r^\star, P^\star) - \mathbb{E}_{\tau \sim \mu_{\text{ref}}}[r^\star(\tau)] \right) \right.$$

$$- \left( J(\pi_{\text{tar}}; r_{\pi_{\text{tar}}}^{\inf}, P^\star) - \mathbb{E}_{\tau \sim \mu_{\text{ref}}}[r_{\pi_{\text{tar}}}^{\inf}(\tau)] \right) \right)$$

$$+ \left( \left( J(\pi_{\text{tar}}; r_{\pi_{\text{tar}}}^{\inf}, P^\star) - \mathbb{E}_{\tau \sim \mu_{\text{ref}}}[r_{\pi_{\text{tar}}}^{\inf}(\tau)] \right) \right.$$

$$- \left( J(\pi_{\text{tar}}; r_{\pi_{\text{tar}}}^{\inf}, P_{\pi_{\text{tar}}}^{\inf}) - \mathbb{E}_{\tau \sim \mu_{\text{ref}}}[r_{\pi_{\text{tar}}}^{\inf}(\tau)] \right) \right)$$

$$+ \left( \left( J(\pi_{\text{tar}}; r_{\pi_{\text{tar}}}^{\inf}, P_{\pi_{\text{tar}}}^{\inf}) - \mathbb{E}_{\tau \sim \mu_{\text{ref}}}[r_{\pi_{\text{tar}}}^{\inf}(\tau)] \right) \right.$$

$$- \left( J(\widehat{\pi}; r_{\widehat{\pi}}^{\inf}, P_{\widehat{\pi}}^{\inf}) - \mathbb{E}_{\tau \sim \mu_{\text{ref}}}[r_{\widehat{\pi}}^{\inf}(\tau)] \right) \right)$$

$$+ \left( \left( J(\widehat{\pi}; r_{\widehat{\pi}}^{\inf}, P_{\widehat{\pi}}^{\inf}) - \mathbb{E}_{\tau \sim \mu_{\text{ref}}}[r_{\widehat{\pi}}^{\inf}(\tau)] \right) \right.$$

$$- \left( J(\widehat{\pi}; r^\star, P^\star) - \mathbb{E}_{\tau \sim \mu_{\text{ref}}}[r^\star(\tau)] \right) \right)$$

$$\leq \left( \left( J(\pi_{\text{tar}}; r^\star, P^\star) - \mathbb{E}_{\tau \sim \mu_{\text{ref}}}[r^\star(\tau)] \right) \right.$$

$$- \left( J(\pi_{\text{tar}}; r_{\pi_{\text{tar}}}^{\inf}, P^\star) - \mathbb{E}_{\tau \sim \mu_{\text{ref}}}[r_{\pi_{\text{tar}}}^{\inf}(\tau)] \right) \right)$$

$$+ \left( \left( J(\pi_{\text{tar}}; r_{\pi_{\text{tar}}}^{\inf}, P^\star) - \mathbb{E}_{\tau \sim \mu_{\text{ref}}}[r_{\pi_{\text{tar}}}^{\inf}(\tau)] \right) \right.$$

$$- \left( J(\pi_{\text{tar}}; r_{\pi_{\text{tar}}}^{\inf}, P_{\pi_{\text{tar}}}^{\inf}) - \mathbb{E}_{\tau \sim \mu_{\text{ref}}}[r_{\pi_{\text{tar}}}^{\inf}(\tau)] \right) \right)$$

$$+ \left( \left( J(\widehat{\pi}; r_{\widehat{\pi}}^{\inf}, P_{\widehat{\pi}}^{\inf}) - \mathbb{E}_{\tau \sim \mu_{\text{ref}}}[r_{\widehat{\pi}}^{\inf}(\tau)] \right) \right.$$

$$- \left( J(\widehat{\pi}; r^\star, P^\star) - \mathbb{E}_{\tau \sim \mu_{\text{ref}}}[r^\star(\tau)] \right) \right)$$

$$\leq \left( \left( J(\pi_{\text{tar}}; r^\star, P^\star) - \mathbb{E}_{\tau \sim \mu_{\text{ref}}}[r^\star(\tau)] \right) \right.$$

$$- \left( J(\pi_{\text{tar}}; r_{\pi_{\text{tar}}}^{\inf}, P^\star) - \mathbb{E}_{\tau \sim \mu_{\text{ref}}}[r_{\pi_{\text{tar}}}^{\inf}(\tau)] \right) \right)$$

$$+ \left( \left( J(\pi_{\text{tar}}; r_{\pi_{\text{tar}}}^{\inf}, P^\star) - \mathbb{E}_{\tau \sim \mu_{\text{ref}}}[r_{\pi_{\text{tar}}}^{\inf}(\tau)] \right) \right.$$

$$- \left( J(\pi_{\text{tar}}; r_{\pi_{\text{tar}}}^{\inf}, P_{\pi_{\text{tar}}}^{\inf}) - \mathbb{E}_{\tau \sim \mu_{\text{ref}}}[r_{\pi_{\text{tar}}}^{\inf}(\tau)] \right) \right)$$

$$\leq \sqrt{\frac{c C_r^2(\mathcal{G}_r, \pi_{\text{tar}}, \mu_{\text{ref}}) \kappa^2 \log(\mathcal{N}_{\mathcal{G}_r}(1/N)/\delta)}{N}}$$

$$+ H r_{\max} \sqrt{\frac{c C_P^2(\{\mathcal{G}_{P_h}\}, \pi_{\text{tar}}) \log(H \mathcal{N}_P(1/N)/\delta)}{N}},$$

where the third and fourth step are due to the definition of $\widehat{\pi}, (r_{\widehat{\pi}}^{\inf}, P_{\widehat{\pi}}^{\inf})$ and (1) in Lemma 3. The last step comes from Lemma 4 and the proof of Theorem 1. This concludes our proof.

### F.1 Proof of Lemma 4

Let $P^h$ be the system dynamics $(P_0^\star, \{P_t^\star\}_{t=1}^h, \{P_t\}_{t=h+1}^{H-1})$ for all $0 \leq h \leq H-1$. Then we have

$$J(\pi_{\text{tar}}; r, P^\star) - J(\pi_{\text{tar}}; r, P)$$

$$= \sum_{h=1}^{H-1} (J(\pi_{\text{tar}}; r, P^h) - J(\pi_{\text{tar}}; r, P^{h-1}))$$

$$+ (J(\pi_{\text{tar}}; r, P^0) - J(\pi_{\text{tar}}; r, P)).$$

For any $h \in [H-1]$, we have

$$J(\pi_{\text{tar}}; r, P^h) - J(\pi_{\text{tar}}; r, P^{h-1})$$

$$= \mathbb{E}_{(s_1, a_1, \cdots, s_h, a_h) \sim (\pi_{\text{tar}}, P^\star)} \Big[ \sum_{s_{h+1}} P_h^\star(s_{h+1}|s_h, a_h) \mathbb{E}_{(\pi_{\text{tar}}, P)}$$

$$[r(\tau)|s_1, a_1, \cdots, s_{h+1}] - \sum_{s_{h+1}} P_h(s_{h+1}|s_h, a_h) \mathbb{E}_{(\pi_{\text{tar}}, P)}$$

$$[r(\tau)|s_1, a_1, \cdots, s_{h+1}] \Big]$$

$$=\mathbb{E}_{(s_1,a_1,\cdots,s_h,a_h)\sim(\pi_{\mathrm{tar}},P^\star)}\Big[\sum_{s_{h+1}}(P_h^\star(s_{h+1}|s_h,a_h)$$

$$-P_h(s_{h+1}|s_h,a_h))\mathbb{E}_{(\pi_{\mathrm{tar}},P)}\big[r(\tau)|s_1,a_1,\cdots,s_{h+1}\big]\Big]$$

$$\leq r_{\max}\mathbb{E}_{(s_h,a_h)\sim(\pi_{\mathrm{tar}},P^\star)}\Big[\big\|P_h^\star(\cdot|s_h,a_h)-P_h(\cdot|s_h,a_h)\big\|_1\Big]$$

$$\leq r_{\max}\sqrt{\frac{cC_P^2(\pi_{\mathrm{tar}})\log(H\mathcal{N}_{\mathcal{G}_{P_h}}(1/N)/\delta)}{N}},$$

where $\mathbb{E}_{(\pi_{\mathrm{tar}},P)}\big[\cdot|s_1,a_1,\cdots,s_{h+1}\big]$ is the distribution of the trajectory $\tau$ when executing policy $\pi_{\mathrm{tar}}$ under the transition probability $\{P_t\}_{t=h+1}^{H-1}$ while fixing the history to be $s_1,a_1,\cdots,s_{h+1}$. Here the first step utilizes the Tower property, the third and fourth step uses Cuachy-Schwartz inequality and the last step comes from Lemma 3.

For $J(\pi_{\mathrm{tar}};r,P^0)-J(\pi_{\mathrm{tar}};r,P)$, similarly we have

$$J(\pi_{\mathrm{tar}};r,P^0)-J(\pi_{\mathrm{tar}};r,P)$$

$$\leq r_{\max}\sqrt{\frac{cC_P^2(\pi_{\mathrm{tar}})\log(H\mathcal{N}_{\mathcal{G}_{P_0}}(1/N)/\delta)}{N}}.$$

Therefore we conclude that

$$J(\pi_{\mathrm{tar}};r,P^\star)-J(\pi_{\mathrm{tar}};r,P)$$

$$\leq Hr_{\max}\sqrt{\frac{cC_P^2(\pi_{\mathrm{tar}})\log(H\mathcal{N}_P(1/N)/\delta)}{N}}.$$

# G   Proof of Theorem 5

We first derive the guarantee of MLE for estimating $A^\star$. Similar to Lemma 1 and Lemma 2, we have the following lemma in the action-based comparison setting:

**Lemma 5.** *Under Assumption 7, with probability at least $1-\delta$, the following event holds true: $\forall h\in[H]$,*

$$\mathbb{E}_{s\sim\mu_h,a^0\sim\mu_{0,h}(\cdot|s),a^1\sim\mu_{1,h}(\cdot|s)}\Big[\Big\|P_{\widehat{A}_h}(\cdot|s,a^0,a^1)$$

$$-P_{A_h^\star}(\cdot|s,a^0,a^1)\Big\|_1^2\Big]\leq\frac{c\log(H\mathcal{N}_{\mathcal{G}_{A_h}}(1/N)/\delta)}{N}.$$

The proof is omitted here. Let $\mathcal{E}_4$ denote the event in Lemma 5. Then under Assumption 8, we can apply the mean value theorem and obtain that under $\mathcal{E}_4$, we have for all $h\in[H]$ that

$$\mathbb{E}_{s\sim\mu_h,a^0\sim\mu_{0,h}(\cdot|s),a^1\sim\mu_{1,h}(\cdot|s)}\Big[|A_h^\star(s,a^0)-A_h^\star(s,a^1)$$

$$-\widehat{A}_h(s,a^0)+\widehat{A}_h(s,a^1)|^2\Big]$$

$$\leq\frac{c\kappa^2\log(H\mathcal{N}_{\mathcal{G}_{A_h}}(1/N)/\delta)}{N},\forall h\in[H]. \tag{6}$$

Recall that $\kappa=\frac{1}{\inf_{x\in[-r_{\max},r_{\max}]}\Phi'(x)}$.

On the other hand, note that we have the following performance lemma:

**Lemma 6.** *For any deterministic Markovian policies $\pi$ and $\pi'$, we have*

$$J(\pi;r^\star,P^\star)-J(\pi';r^\star,P^\star)$$

$$=\sum_{h=1}^{H}\mathbb{E}_{s\sim d_h^{\pi'}}\Big[Q_h^\pi(s,\pi(s))-Q_h^\pi(s,\pi'(s))\Big]$$

The proof is deferred to Appendix G.1.

The rest of the proof largely follows Uehara et al. (2023). Under the event $\mathcal{E}_4$, we can bound the suboptimality of $\widehat{\pi}$ as follows:

$$J(\pi^\star;r^\star,P^\star)-J(\widehat{\pi};r^\star,P^\star)$$

$$\leq r_{\max}\sum_{h=1}^{H}\mathbb{E}_{s\sim d_h^{\pi^\star}}\Big[\mathbb{1}(\pi_h^\star(s)\neq\widehat{\pi}_h(s))$$

$$\cdot\mathbb{1}(Q_h^\star(s,\widehat{\pi}_h(s))<Q_h^\star(s,\pi_h^\star(s)))\Big]$$

$$\leq r_{\max}\sum_{h=1}^{H}\mathbb{E}_{s\sim d_h^{\pi^\star}}\Big[\sum_{a\in\mathcal{A}}\mathbb{1}\Big(\widehat{A}_h(s,a)\geq\widehat{A}_h(s,\pi_h^\star(s))\Big)$$

$$\cdot\mathbb{1}\Big(Q_h^\star(s,a)<Q_h^\star(s,\pi_h^\star(s))\Big)\Big],$$

where the first step comes from Lemma 6 and the second step is due to the definition of $\widehat{\pi}$. Then for any $\alpha>0$, we have

$$\mathbb{E}_{s\sim d_h^{\pi^\star}}\Big[\sum_{a\in\mathcal{A}}\mathbb{1}\Big(\widehat{A}_h(s,a)\geq\widehat{A}_h(s,\pi_h^\star(s))\Big)$$

$$\cdot\mathbb{1}\Big(Q_h^\star(s,a)<Q_h^\star(s,\pi_h^\star(s))\Big)\Big]$$

$$\leq\mathbb{E}_{s\sim d_h^{\pi^\star}}\Big[\sum_{a\in\mathcal{A}}\mathbb{1}\Big(Q_h^\star(s,\pi_h^\star(s))>Q_h^\star(s,a)\geq Q_h^\star(s,\pi_h^\star(s))-\alpha\Big)\Big]$$

$$+\mathbb{E}_{s\sim d_h^{\pi^\star}}\Big[\sum_{a\in\mathcal{A}}\mathbb{1}\Big(Q_h^\star(s,\pi_h^\star(s))-Q_h^\star(s,a)$$

$$-\widehat{A}_h(s,\pi_h^\star(s))+\widehat{A}_h(s,a)\geq\alpha\Big)\Big].$$

By Assumption 6, we have

$$\mathbb{E}_{s\sim d_h^{\pi^\star}}\Big[\sum_{a\in\mathcal{A}}\mathbb{1}\Big(Q_h^\star(s,\pi_h^\star(s))>Q_h^\star(s,a)\geq Q_h^\star(s,\pi_h^\star(s))-\alpha\Big)\Big]$$

$$\leq|\mathcal{A}|(\alpha/\alpha_0)^\beta.$$

For the second term, we have

$$\mathbb{E}_{s\sim d_h^{\pi^\star}}\Big[\sum_{a\in\mathcal{A}}\mathbb{1}\Big(Q_h^\star(s,\pi_h^\star(s))-Q_h^\star(s,a)$$

$$- \widehat{A}_h(s, \pi_h^\star(s)) + \widehat{A}_h(s, a) \geq \alpha \Big) \Big]$$

$$= \frac{1}{\alpha^2} \mathbb{E}_{s \sim d_h^{\pi^\star}} \left[ \sum_{a \in \mathcal{A}} \alpha^2 \mathbb{1} \Big( A_h^\star(s, \pi_h^\star(s)) - A_h^\star(s, a) \right.$$

$$\left. - \widehat{A}_h(s, \pi_h^\star(s)) + \widehat{A}_h(s, a) \geq \alpha \Big) \right]$$

$$\leq \frac{1}{\alpha^2} \mathbb{E}_{s \sim d_h^{\pi^\star}} \left[ \sum_{a \in \mathcal{A}} \left| A_h^\star(s, \pi_h^\star(s)) - A_h^\star(s, a) \right.\right.$$

$$\left.\left. - \widehat{A}_h(s, \pi_h^\star(s)) + \widehat{A}_h(s, a) \right|^2 \right]$$

$$\leq \frac{c|\mathcal{A}|C_{\mathrm{act}}\kappa^2 \log(H\mathcal{N}_{\mathcal{G}_{A_h}}(1/N)/\delta)}{\alpha^2 N},$$

where the last step comes from the definition of $C_{\mathrm{act}}$ and (6).

Therefore by picking appropriate $\alpha$, we have with probability at least $1 - \delta$ that

$$J(\pi^\star; r^\star, P^\star) - J(\widehat{\pi}; r^\star, P^\star) \leq cH|\mathcal{A}| \left(\frac{2}{\beta}\right)^{\frac{\beta-2}{\beta+2}} \left(\frac{1}{\alpha_0}\right)^{\frac{2\beta}{\beta+2}}$$

$$\cdot \left(\frac{\kappa^2 C_{\mathrm{act}} \log(H\mathcal{N}_{\mathcal{G}_A}(1/N)/\delta)}{N}\right)^{\frac{\beta}{\beta+2}}.$$

### G.1 Proof of Lemma 6

For any two policies $\pi$ and $\pi'$, we have that

$$J(\pi'; r^\star, P^\star) - J(\pi; r^\star, P^\star)$$

$$= \mathbb{E}_{\pi'} \left[ r_1^\star(s_1, a_1) + V_2^{\pi'}(s_2) \right] - \mathbb{E}_{\pi'} \left[ V_1^\pi(s_1) \right]$$

$$= \mathbb{E}_{\pi'} \left[ V_2^{\pi'}(s_2) - (V_1^\pi(s_1) - r_1^\star(s_1, a_1)) \right]$$

$$= \mathbb{E}_{\pi'} \left[ V_2^{\pi'}(s_2) - V_2^\pi(s_2) \right] + \mathbb{E}_{\pi'} \left[ Q_1^\pi(s_1, a_1) - V_1^{r,\pi}(s_1) \right]$$

$$= \mathbb{E}_{\pi'} \left[ V_2^{\pi'}(s_2) - V_2^\pi(s_2) \right] + \mathbb{E}_{\pi'} \left[ \langle Q_1^\pi(s_1, \cdot), \pi_1'(\cdot|s_1) - \pi_1(\cdot|s_1) \rangle \right]$$

$$= \cdots = \sum_{h=1}^H \mathbb{E}_{\pi'} \left[ \langle Q_h^\pi(s_h, \cdot), \pi_h'(\cdot|s) - \pi_h(\cdot|s) \rangle \right].$$

This concludes our proof.

