# OpenReview forum: "Provable Offline Reinforcement Learning with Human Feedback"
_ICML.cc/2023/Workshop/ILHF — ILHF Workshop ICML 2023_

### Official Review · Reviewer_jveo · 2023-06-14

**Rating:** 7
**Confidence:** 2

**Review:**

This paper introduces algorithms and analysis for preference-based RL with human feedback (RLHF) from offline data. This is an important direction because these methods would better be able to leverage previously collected data to improve policies through human feedback from just preferences over trajectories. They prove that the algorithm can approach the performance of a target policy given sufficient dataset coverage when the transition kernel is known, and also provide analysis in the case where transition dynamics are not known. Finally they consider the setting where preferences are defined not over entire trajectories but instead on individual actions.


Strengths:
- The authors do well at explaining the differences between their work and prior work (Zhu et al. 2023), and their work is more generalized (general function approximation vs linear models, consideration of action-based preferences) which is a clear strength.
- The problem is important and timely, and is a good fit for this workshop.
- The arguments and findings in the paper are presented clearly. While I did not carefully review the proofs, the text effectively described and contextualized the method. The algorithms in the paper appear to significantly advance existing methods for provable offline RLHF.

Weaknesses:
- Minor point: the references to the algorithm boxes in the text seem to point to the incorrect lines, for instance, in Section 4.1, both the confidence set construction and distributionally robust policy optimization sections point to Line 1. This also occurs in Section 6.3.

---

### Official Review · Reviewer_KzYW · 2023-06-16
**Good paper proposing and analyzing offline RL algorithms using preferences instead of rewards, though the practicality of the results may be limited.**

**Rating:** 7
**Confidence:** 3

**Review:**

Review:
---

The paper proposes and analyzes offline RL algorithms using preference-based data, whether over trajectories or actions. The algorithm learns a MLE estimate of the reward function using the preferences, constructs confidence intervals around the MLE reward, then minimizes a pessimistic policy in the worst-case around the possible reward functions. The authors show that the algorithm has sample complexity comparable to existing offline RL algorithms that use the reward.

Overall, I think the paper makes a valuable contribution given the popularity of preference-based learning. I am concerned on how practical the proposed algorithm is. For one, the paper assume a unique MLE reward function can be learned whereas in practice, multiple may exist depending on the parameterization of the reward function. Second, the algorithm involves solving a max-min problem to optimize a policy in the worst-case over the confidence set of rewards. It would be beneficial if the paper could explain how this can be done feasibly for general function classes.

Finally, there appear to be some typos in the submitted manuscript. For instance, in describing the algorithm the authors reference line numbers that do not seem consistent with the actual algorithm. Second, the margins for certain equations seem to overflow into the text (e.g. Assumption 1).

---

### Decision · Program_Chairs · 2023-06-20

Accept